# Stable skyrmion bundles at room temperature and zero magnetic field in a chiral magnet

Yongsen Zhang[1,2,6], Jin Tang [3,6] ✉, Yaodong Wu[4], Meng Shi[1,2], Xitong Xu [2], Shouguo Wang [5], Mingliang Tian [2,3] & Haifeng Du [2] ✉

Topological spin textures are characterized by magnetic topological charges, $Q$, which govern their electromagnetic properties. Recent studies have achieved skyrmion bundles with arbitrary integer values of $Q$, opening possibilities for exploring topological spintronics based on $Q$. However, the realization of stable skyrmion bundles in chiral magnets at room temperature and zero magnetic field − the prerequisite for realistic device applications − has remained elusive. Here, through the combination of pulsed currents and reversed magnetic fields, we experimentally achieve skyrmion bundles with different integer $Q$ values − reaching a maximum of 24 at above room temperature and zero magnetic field − in the chiral magnet $Co_8Zn_{10}Mn_2$. We demonstrate the field-driven annihilation of high-$Q$ bundles and present a phase diagram as a function of temperature and field. Our experimental findings are consistently corroborated by micromagnetic simulations, which reveal the nature of the skyrmion bundle as that of skyrmion tubes encircled by a fractional Hopfion.

Magnetic skyrmions are vortex-like spin textures characterized by an integer magnetic topological charge[1–3], defined as

$$Q = \frac{1}{4\pi} \int \mathbf{m} \cdot \left( \frac{\partial \mathbf{m}}{\partial x} \times \frac{\partial \mathbf{m}}{\partial y} \right) dxdy. \qquad (1)$$

Magnetic topological charge $Q$ plays a crucial role in determining various topology-related properties of skyrmions, including skyrmion Hall effects[4,5], topological Hall effects[6], ultrasmall depinning current[7], particle-like physics[8], and electric transport properties[9,10]. Despite the significance of $Q$ in determining electromagnetic properties of topological spin textures[11–19], traditional skyrmions are constrained to possess a fixed value of $|Q| = 1$. Recent studies propose a strategy to extend the values of $Q$ from 1 to any integers in an assembly, called

skyrmion bundles or skyrmion bags[20–22]. Skyrmion bundles consist of various skyrmions encircled by a closed spin spiral. The $N$ internal skyrmions contribute $Q = N$, while the outer spin spiral contributes $Q = -1$. As a result, skyrmion bundles possess a total topological charge of $Q = N - 1$. Skyrmion bundles, holding the advantages of diversity $Q$ and morphologies, have greatly enriched the family of topological magnetic solitons and shown potential application in extended topological spintronic devices, such as binary racetrack memories using any two types of skyrmion bundles, multiple-bits ASCII information encoding, and interconnect devices[23–28].

The realization of skyrmion bundles is based on the first-order magnetic phase change from the helix to the skyrmion lattice[29], which allows for the coexistence of the two phases[30,31]. Skyrmion bundles can be created by applying reversed magnetic fields to the coexisting

[1]Science Island Branch, Graduate School of USTC, Hefei 230026, China. [2]Anhui Province Key Laboratory of Low-Energy Quantum Materials and Devices, High Magnetic Field Laboratory, HFIPS, Chinese Academy of Sciences, Hefei 230031, China. [3]School of Physics and Optoelectronic Engineering, Anhui University, Hefei 230601, China. [4]School of Physics and Materials Engineering, Hefei Normal University, Hefei 230601, China. [5]Anhui Key Laboratory of Magnetic Functional Materials and Devices, School of Materials Science and Engineering, Anhui University, Hefei 230601, China. [6]These authors contributed equally: Yongsen Zhang, Jin Tang. ✉e-mail: jintang@ahu.edu.cn; duhf@hmfl.ac.cn

skyrmion-helix phases[20]. The formation of coexisting skyrmion-helix phases typically involves a complex operation, often through a cooling procedure from high temperatures[20,32]. Furthermore, the observation of skyrmion bundles in chiral magnets has been limited to temperatures far below room temperature and under a certain magnetic field[32], which poses practical challenges for their application in real-world devices.

In this work, we successfully observed isolated skyrmion bundles with varying integer $Q$ values, including a maximum of 24, and reported the unambiguous experimental realization at room temperature and zero magnetic field in $\beta$-Mn-type $Co_8Zn_{10}Mn_2$ chiral magnet by the combination of pulsed currents and reversed magnetic fields. The creation of room-temperature skyrmion bundles avoids the cooling procedure. Furthermore, we demonstrate the field-driven topological phase transition and provide a stabilization diagram of skyrmion bundles as a function of magnetic field and temperatures. Finally, we further elucidated the connection between skyrmion bundles and magnetic Hopfions[33–36].

## Results

### Observation of skyrmion bundles at room temperature without a field cooling procedure

Figure 1a shows the scenario of creating skyrmion bundles at zero magnetic fields, which evolves 3 steps: we first achieve conical-to-skyrmion transformations by applying pulsed currents at negative magnetic fields, then we realize skyrmion bundles by applying

reversed magnetic fields, skyrmion bundles can maintain its stability even the magnetic field is reduced to zero. Skyrmion bundles in chiral magnets consist of an interior skyrmion bag and superficial multi-$Q$ chiral vortices, as shown in Fig. 1b–g. Fig. 1 contains a representative simulated 3D magnetic configuration of skyrmion bundles containing 3 skyrmions. The iso-surfaces correspond to a value of −0.1 for the normalized out-of-plane magnetization component, i.e., $m_z = -0.1$ (Fig. 1b, d). When approaching the sample surfaces (Fig. 1e, g), where the magnetic vortex is a bi-antiskyrmion with $Q = 2$, and the complete skyrmion bags with $Q = 2$ are located only in the middle layers (Fig. 1f). Therefore, despite the strong spin twist along the depth dimension due to the conical background magnetizations, topological charges maintain $Q = 2$ throughout all layers (Supplementary Fig. 1 and Movie 1). Noted that the smooth spin modulations between skyrmion bags in interior layers and antiskyrmions in surface layers without topological variations do not necessarily the emergence of Bloch points, which is different from quadrupole of Bloch points sewing skyrmions and antiskyrmions with topological reversals[37,38].

The skyrmions inside the bundle exhibit conventional characteristics, with polarity $p = 1$ and vorticity $v = 1$, resulting in a topological charge ($Q$) of 1 for each skyrmion[39–41]. The peripheral helical stripe possesses the same vorticity but opposite polarity. Consequently, the topological charge of skyrmion bundles is the summation of the central skyrmions and the boundary helical stripe, expressed as $Q = N - 1$, where $N$ represents the number of skyrmions within the bundle. Given the fact for the first-order transition from

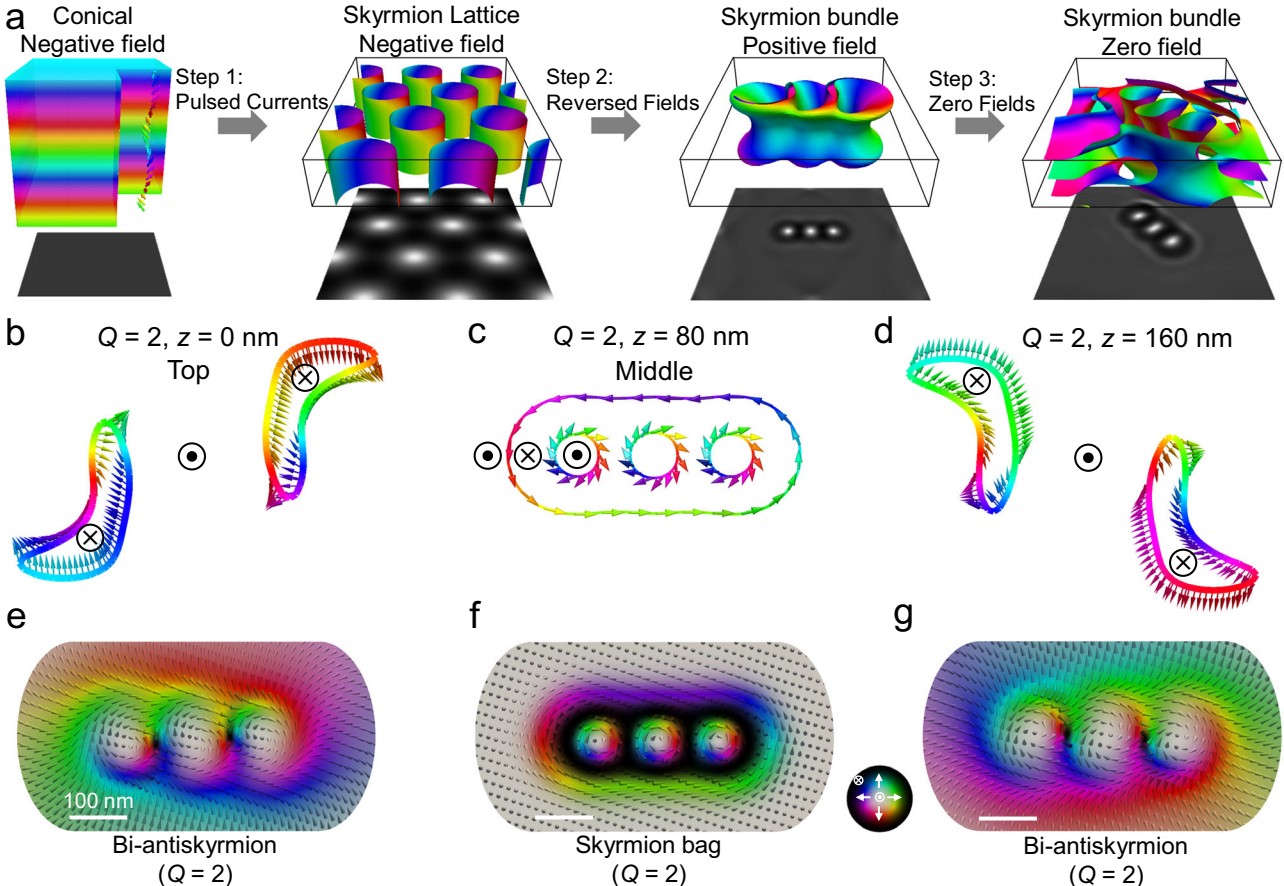

**Fig. 1 | Scenario of creating zero-field skyrmion bundles and their three-dimensional magnetic configurations. a** Scenario of creating skyrmion bundles. The images below display corresponding simulated Fresnel images. **b**−**d** Contour of $m_z = -0.1$ at the top surface ($z = 0$ nm), at the middle layer ($z = 80$ nm), and the bottom surface ($z = 160$ nm) of a $Q = 2$ bundle. ⊙ and ⊗ represent up and down orientations of the polarity. **e**−**g** Magnetic configurations at the top, middle, and bottom layers of the $Q = 2$ bundles. The colorwheel represents the magnetizations.

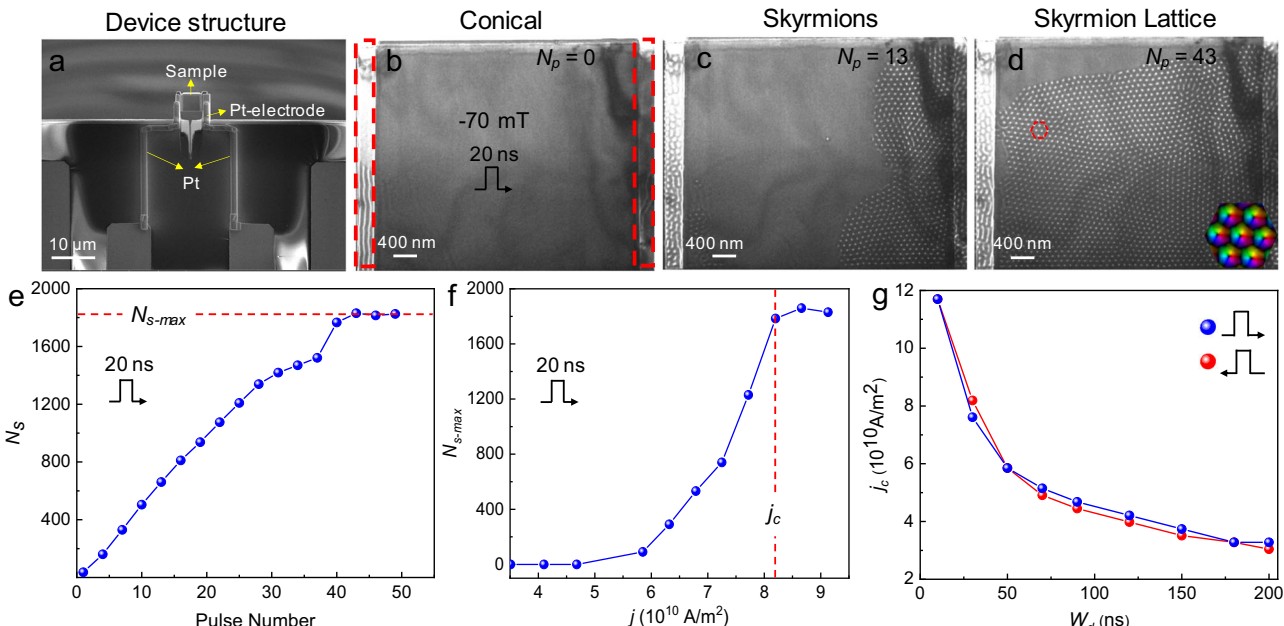

**Fig. 2 | Current-induced creation of skyrmions at room temperature. a** The image of device structure obtained from scanning electron microscopy imaging. **b–d** A sequence of Fresnel images of the skyrmion creation process after applying numbers of the current pulse, corresponding to conical, skyrmions, and skyrmion lattice, respectively. The red rectangular dashed lines are the thin area on both sides of the sample in (**b**). Inset: in-plane magnetization mapping of skyrmion lattice within a hexagonal dashed line by Transport of Intensity Equation (TIE). **e** Skyrmion number $N_s$ as a function of pulse numbers, the pulse width is 20 ns. **f** Maximum skyrmion number $N_{s\text{-max}}$ as a function of the current density, the pulse width is 20 ns. **g** Threshold current density $j_c$ required to achieve the maximum skyrmion number at different pulse width $W_d$. The data points in **e** are taken from Supplementary Movie 2. Defocused distance, −1200 μm.

helix to skyrmions, the coexistence of skyrmions and helix is allowed and serves as a precursor to the formation of skyrmion bundles. We first demonstrate the creation of skyrmions induced by currents[29,42]. The micro-device utilized in the experiment comprises two Pt electrodes and a ~160 nm thick lamella with two narrow regions of ~80 nm thickness on both sides, fabricated from $Co_8Zn_{10}Mn_2$ that is a bulk chiral magnet with DMI-stabilized skyrmions (Fig. 2a and Supplementary Fig. 2).

Figure 2 illustrates the process of skyrmion creation through the application of a series of current pulses with varying pulse widths and current densities in the x-direction. Initially, as shown in Fig. 2b, we obtained a conical state shown by uniform Fresnel contrasts at $B = -70$ mT. Subsequently, upon applying current pulses with a duration of 20 ns and a density of $8.19 \times 10^{10}$ A/m², skyrmion clusters were generated after several pulses (Fig. 2c). As the number of pulses increased, the skyrmion clusters gradually merged to form the skyrmion lattice, as depicted in Fig. 2d. Figure 2e shows the effect of the current pulse number on the skyrmion creation under a current density $j \sim 8.42 \times 10^{10}$ A/m². Initially, the number $N_s$ of skyrmions exhibits a positive linear relationship with the number of pulses $N_p$. After approximately the application of 43 pulsed currents, skyrmions covered most regions of the sample, with a total count of $N_s \sim 1830$. Subsequently, the number of skyrmions almost keeps a balance around ~1830, which is defined as the maximum skyrmion number $N_{s\text{-max}}$, despite the further application of more pulsed currents (see Supplementary Fig. 3 for details). Figure 2f shows the dependence of skyrmion maximum numbers ($N_{s\text{-max}}$) on the current density when the pulse width was set to 20 ns. Firstly, as the current density increases, the number of skyrmions rises until it reaches a maximum value. However, once the current density exceeds a certain threshold, the current density no longer rises, as excessively high current densities could damage the sample[30,43]. Figure 2g displays the effect of pulse width on the critical current densities and directions. Despite the opposite current directions, the difference in critical current density is relatively small under the same pulse width.

Skyrmions are first created at the edge of the sample and then pushed into the center[44]. This creation process can be understood by the combined current effect of spin transfer torque and Joule thermal heating[23,45,46], as shown in Supplementary Movie 2. Because the current density is inversely proportional to the thickness, skyrmions are first created at the thin region because of larger temperature increases induced by the current. Then, the spin transfer torque could drive skyrmions nucleated in the thin region to the thick region. A previous study has shown the current-driven skyrmion-to-bobber transformations in stepped geometry[47]. However, magnetic bobbers are not observed in our experiments (Supplementary Movie 2). The step edge between thin and thick regions in realistic experiments cannot be very sharp. Our simulations show that a slight continuous deformation of the step edge can lead to the transformation of short skyrmion tubes in a thin region to long tubes in a thick region (Supplementary Movie 3), which explains the expansion of skyrmions from the thin region to the thick region.

According to the scenario for creating skyrmion bundles (Fig. 1a), we have shown the creation of skyrmions with positive Q using pulsed current at room temperature without additional field cooling process (Fig. 3a). Additionally, after the pulse current was turned off, positive magnetic fields were applied to the skyrmions. Although $Q = 1$ skyrmions are not thermodynamically stable under positive fields, they can usually survive in the non-equilibrium metastable state (Fig. 3a, 0 and 58 mT) at temperatures far below Curie temperature $T_c$ (~ 350 K). When the magnetic field increased to $B = 80$ mT, most of the skyrmions were annihilated, leaving a skyrmion bundle consisting of a few skyrmions encircled by a dark ring. The Transport of Intensity Equation (TIE)[48] analysis was employed to calculate the projected in-plane magnetization distribution of the skyrmion bundle. As shown in Fig. 3a (80 mT), it demonstrated that the surrounding circular spiral with weak contrast has the opposite rotation sense compared to the skyrmions inside. To quantitatively evaluate the fine variations in the phase shift inside and outside the skyrmion

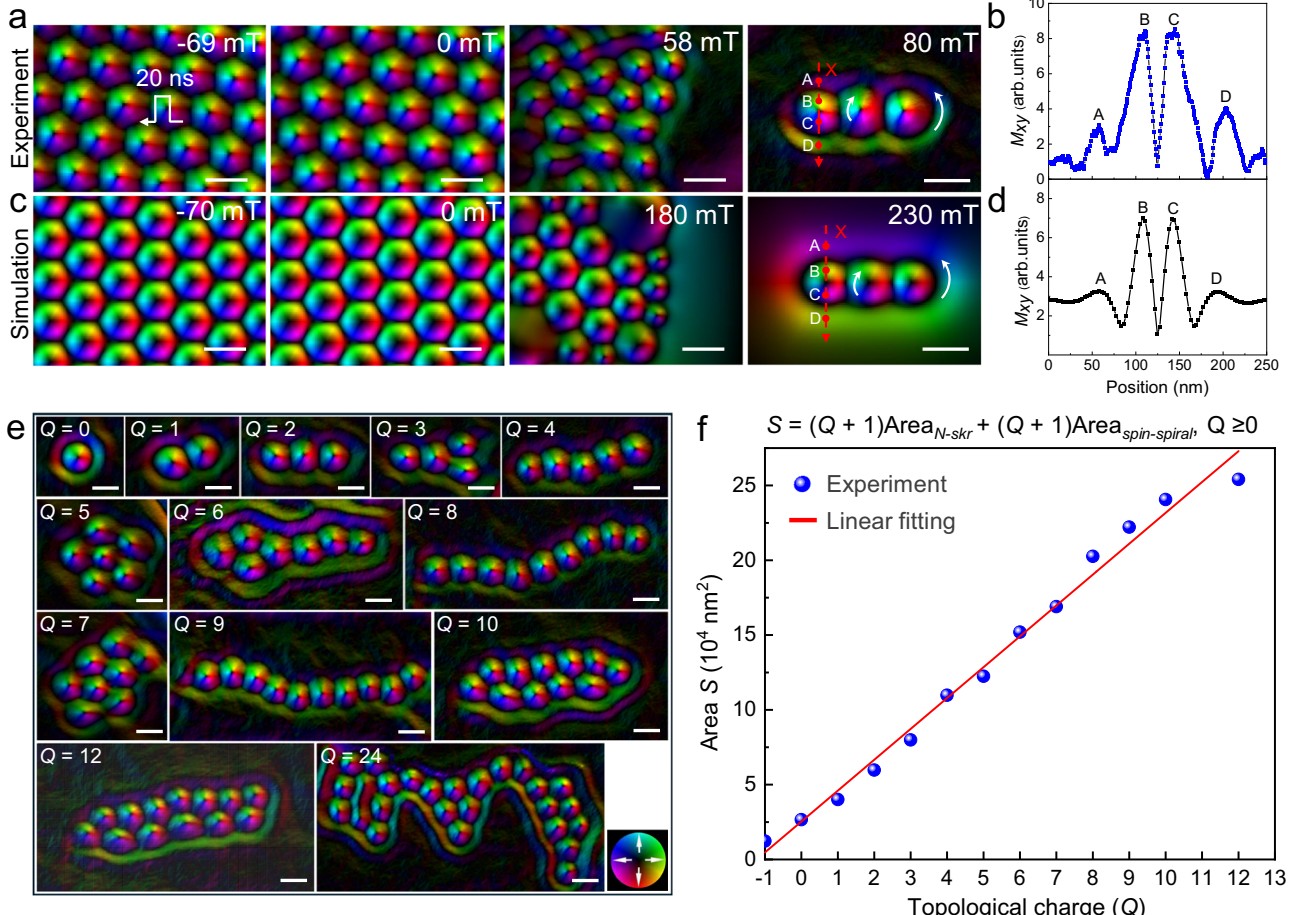

**Fig. 3 | Diversity of room-temperature skyrmion bundles. a** Creation of skyrmion bundles at room temperature by applying reversed positive magnetic fields on skyrmion clusters with positive $Q$. **b** Line profile of magnetic phase shift extracted from (**a**) (red dashed arrow). **c** Corresponding simulated averaged in-plane magnetization mapping during the creation from skyrmion lattice to skyrmion bundles. **d** Line profile of magnetic phase shift extracted from (**c**) (red dashed arrow). **e** In-plane magnetic configurations of representative magnetic skyrmion bundles with varying $Q$ at $B$ ~ 80 mT. **f** Dependence between the area $S$ of different skyrmion bundles and topological charge $Q$. The fitting formula is $S = (Q+1)(S_{N\text{-}skr} + S_{spiral})$, and the data points are taken from bundles presented in (**e**). The colorwheel represents the in-plane magnetizations. Scale bar, 100 nm.

bundle, the line profiles of the experimental and simulated phases were plotted in Figs. 3b and 3d, respectively. It is evident that the contrast intensity of the outer circular spiral is weaker than that of the inner skyrmions. The depth-modulated spin twists (Fig. 1c) contribute weaker magnetic contrasts of the outer spiral compared to those of the interior skyrmions for the average in-plane magnetization mapping of skyrmion bundles. These characteristics of chiral skyrmion bundles in $Co_8Zn_{10}Mn_2$ are highly similar to those in chiral magnet FeGe[20].

By repeating the process of combining pulsed currents and reversed magnetic fields, a diversity of magnetic skyrmion bundles with a maximum $Q$ of 24 at room temperature can be obtained, as shown in Fig. 3e. For the bubble bundles stabilized by magnetic dipole-dipole interaction, the internal skyrmion bubbles are much larger than the size of an isolate skyrmion, leading to the expanded $Q$-related size of bubble bundles[49]. In contrast, the skyrmion bundles stabilized by chiral interactions in $Co_8Zn_{10}Mn_2$ always reveal a compact configuration. The internal skyrmions of the bundles always tightly bind to each other and have a comparable size as that of an isolate skyrmion, leading to a linear increased area $S$ as a function of $Q$. We were able to identify the boundaries of skyrmion bundles and subsequently calculated the area within these determined boundaries (Supplementary Fig. 4), which represents the area occupied by skyrmion bundles. Figure 3f shows the area $S$ of skyrmion bundles as a function of

topological charge $Q$, i.e.,

$$S = (Q+1)(S_{N-skr} + S_{spiral}) \qquad (2)$$

which represents the sum of the area occupied by the inner skyrmions $S_{N-Skr}$ and the outer spin spiral $S_{spiral}$. It should be noted that the combination of pulsed current and reversed magnetic fields can be a promising technique in achieving other fascinating topological solitons, such as the closely bounded skyrmion-antiskyrmion pair[50] (Supplementary Fig. 5), at room temperature.

**Topological phase transition at high and zero magnetic fields**

Figure 4a shows a representative topological phase transition of high-$Q$ skyrmion bundles under high magnetic fields at room temperature. When increasing the external magnetic field from 70 to 79 mT, one interior skyrmion of the $Q = 8$ bundle was annihilated, while the boundary spin spiral remained stable and shrank to tightly bind the remaining skyrmions, resulting in the formation of a $Q = 7$ bundle. As the magnetic field continued to increase, the internal skyrmions gradually disappeared one by one until only one remained, forming a $Q = 0$ bundle, also known as the $2\pi$−vortex or skyrmionium (Fig. 4a, 107 mT)[22,51,52]. The $Q = 0$ bundle could survive in a wide magnetic field range from 107 to 155 mT. However, due to the consistent polarity between the center skyrmion in the $Q = 0$ bundle and the direction of

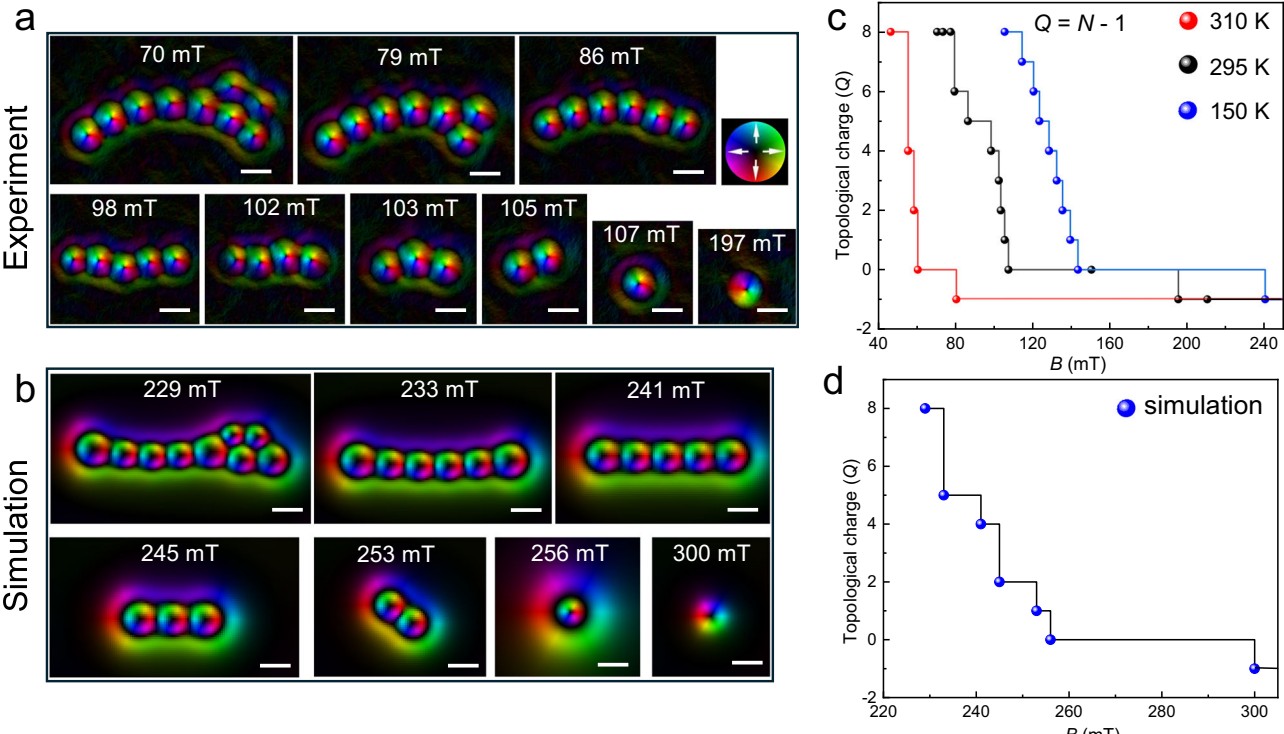

**Fig. 4 | Field-driven quantized topological annihilations. a** Field-driven magnetic evolutions from a skyrmion bundle with $Q = 8$ at 295 K. **b** Corresponding simulated averaged in-plane magnetization mapping during the field-driven magnetic evolution. **c** Topological charge $Q$ as a function of magnetic field $B$. **d** Corresponding simulated topological charge $Q$ as a function of magnetic field $B$. $N$ represents the number of interior skyrmion tubes. The colorwheel represents the in-plane magnetizations. Scale bar, 100 nm.

the positive magnetic field[41,53], the internal skyrmion was inherently unstable and eventually disappeared at $B = 155$ mT. Subsequently, the outer ring shrank to form a skyrmion with $Q = -1$, with a polarity opposite to that of the skyrmion within the bundle. The field-driven topological quantized annihilation can be well reproduced in our zero-temperature simulations (Fig. 4b, d).

We further observed skyrmion bundles in a broad temperature range from 150 to 320 K and explored their field-driven magnetic evolutions, as shown in Fig. 4c. The topological quantized one-by-one annihilations in the field-increasing process work for temperatures below 295 K. The threshold maximum magnetic field required for the stabilization of bundles decreased as the temperature increased. In contrast, at $T \sim 310$ K, the topological quantized annihilation behavior of the $Q = 8$ bundle was not gradual but instead resulted in a transition from 8 to 4, 2, 0, and finally −1. Figure 4d shows the simulated field-driven magnetic evolution from a $Q = 8$ skyrmion bundle at zero temperature. The topological quantized annihilation in the zero-temperature simulation is hardly shown with a continuous one-by-one mode, but instead resulted in a transition from 8 to 5, 4, 2, 1, 0, and finally −1.

The stability of zero field topological spin textures has been typically realized in geometrically confined nanostructures or a compact lattice (Fig. 3a, c)[51,54–56]. It is important to further achieve isolated zero-field topological solitons in free geometries. Here, we show that isolated zero-field skyrmion bundles can be created in the perpendicular helix, whose spins keep uniform within each plane and modulate along the depth to form helix (Supplementary Fig. 8), without the application of geometrical confinement effects. We first show that an isolated skyrmion can be stabilized at zero magnetic fields in a broad temperature range far below the Curie temperature (Supplementary Fig. 6). We then explore the stability of skyrmion bundles in the field-decreasing process (Fig. 5 and Supplementary Fig. 7). When decreasing

the magnetic field to zero from the high-field ferromagnetic state at room temperature, the Fresnel contrasts of the $Co_8Zn_{10}Mn_2$ lamella remain uniform, suggesting the stability of conical or perpendicular helix with zero defocused Fresnel contrasts around zero magnetic fields (Supplementary Fig. 8). Our simulations confirm the formation of the perpendicular helix at zero fields by decreasing field from ferromagnet at high field (Supplementary Fig. 8). Noted that the ferromagnet, conical, and perpendicular helix are all shown no defocused Fresnel contrasts because of zero integral in-plane magnetization component along the depth.

Figure 5a shows the topological quantized annihilation of a $Q = 1$ bundle with a continuous topological quantized annihilation in the field-increasing process. In contrast, the magnetic evolution of the $Q = 1$ bundle in the field-decreasing process is shown in Fig. 5b. The $Q = 1$ skyrmion bundle remains stable even at zero field in the perpendicular helix magnetization background. The $Q = 1$ skyrmion bundle cannot persist and transform to long helix domains until a negative magnetic field $B = -9$ mT is applied (Fig. 5b). By repeating the field-decreasing process, we confirm the stability of skyrmion bundles with other $Q$ values at zero magnetic fields and room temperatures (Fig. 5c and Supplementary Fig. 7c, d). Our simulations well reproduce the stability of skyrmion bundles in the field-decreasing process with constant $Q$ values (Supplementary Fig. 7a, b).

Figure 5d−f depicts the magnetic phase diagram of skyrmion bundles as a function of temperature and magnetic field in a field-decreasing process from an initial $Q = 0$, 1, and 2 bundles. Skyrmion bundles in $Co_8Zn_{10}Mn_2$ magnets can survive in a broad field-temperature region. However, for temperatures above 320 K, strong thermal fluctuations promote the transformation from high-energy metastable skyrmion bundles to helical phases with lower energy (Supplementary Figs. 9 and 11). The metastable skyrmion

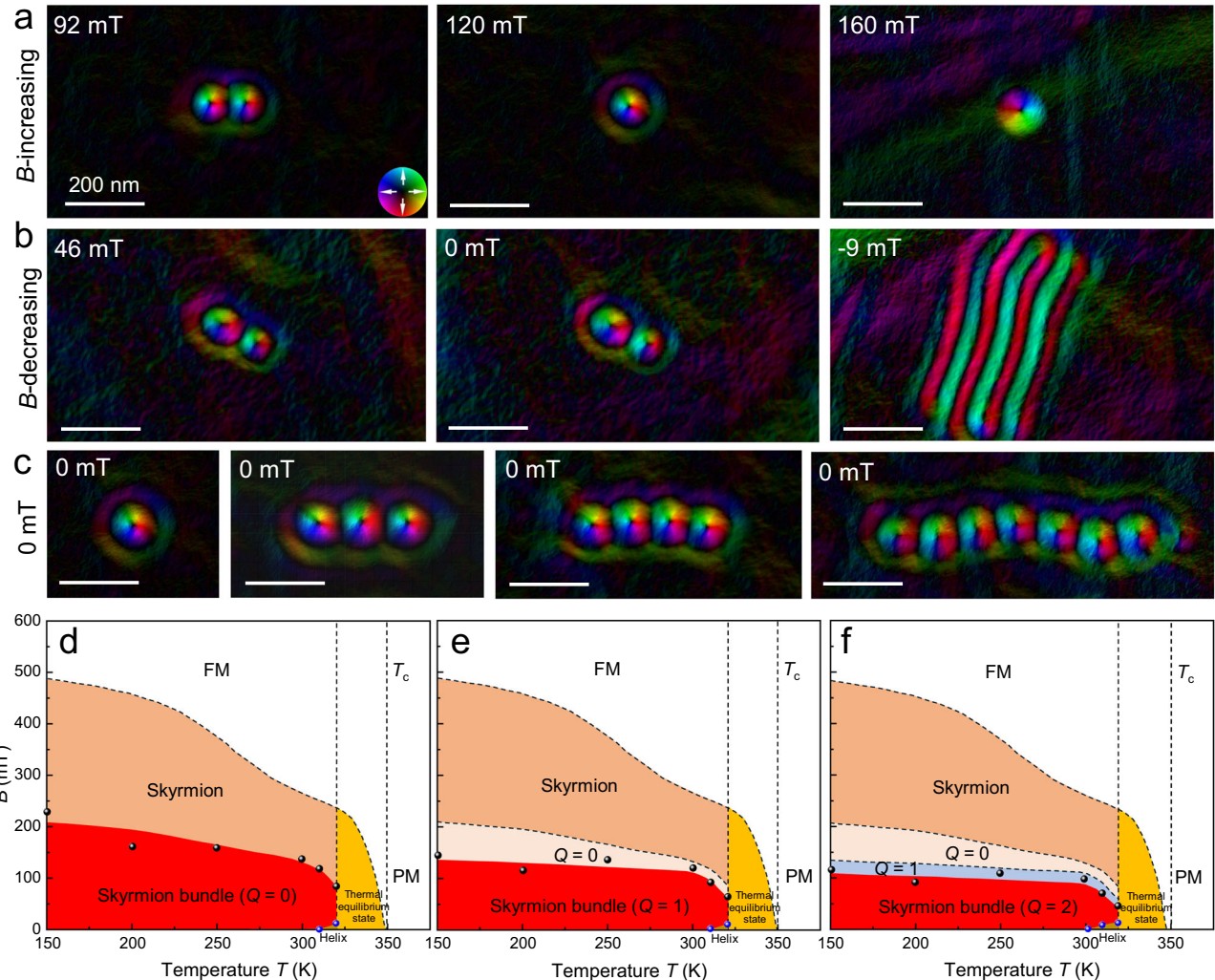

**Fig. 5 | Magnetic phase diagram of skyrmion bundles. a** The topological quantized annihilation of a $Q=1$ bundle with increasing field. **b.** Topological transformation from a $Q=1$ bundle to the magnetic helix in the field-decreasing process. **c** Stable skyrmion bundles under zero magnetic field. **d–f** Magnetic phase diagrams of the skyrmion bundles (0, 1, and 2) as a function of temperature and magnetic field. The data points in the diagram are the critical points of the two-state transition. FM and PM represent ferromagnetic and paramagnetic states, respectively. The colorwheel represents the in-plane magnetization distributions in (**a–c**).

bundles and the helix at low fields are all disturbed in high temperatures near the Curie temperature. When the temperature further increases to 320 K, the zero-field conical state is not stable and transforms to a helix (Supplementary Fig. 9) due to the strong thermal fluctuation energy near the Curie temperature. From initially a $Q=0$ bundle at room temperature and zero field, the skyrmion bundle turns to helix domains at an elevated temperature of 315 K (Supplementary Fig. 11).

**Fractional Hopfion rings in skyrmion bundles**

Magnetic Hopfions are topologically stable, three-dimensional magnetic structures that exhibit unique and fascinating properties[33–36]. These localized spin configurations arise in magnetic materials with nontrivial symmetry, such as chiral magnets or frustrated systems. Unlike conventional magnetic domain walls or skyrmions that have a one-dimensional or two-dimensional structure, respectively, Hopfions possess a complex three-dimensional magnetic texture resembling a toroidal knot. Due to their stability and intriguing characteristics, Hopfions have attracted significant attention in the field of spintronic devices. Understanding the fundamental properties and dynamics of these magnetic structures holds great promise for developing novel technologies in the future. The topological index of Hopfions can be

mathematically represented by the equation

$$Q_H = -\frac{1}{(8\pi)^2}\int \mathbf{F}\cdot\mathbf{A}\mathrm{d}r \tag{3}$$

where $\mathbf{A}$ represents the vector potential of $\mathbf{F}=\nabla\times\mathbf{A}$. The vector $\mathbf{F}$ can be expressed using the local magnetization direction, represented by the unit vector $\mathbf{n}$, and the Levi-Civita permutation symbol $\varepsilon$. Specifically, the components of $F_i$ can be defined as

$$F_i = \varepsilon_{ijk}\mathbf{n}\cdot(\partial_j\mathbf{n}\times\partial_k\mathbf{n}) \tag{4}$$

We then explore the relationship between skyrmion bundles and magnetic Hopfions. Taking $Q=0$ bundles as an example, we show different iso-surfaces for different $m_z$ values, as shown in Fig. 6a. For iso-surfaces of $m_z > 0.6$, the in-plane magnetizations $m_{xy}$ reveal the same characteristics as the magnetic Hopfion. As the Hopfion charge $Q_H$ of the $Q=0$ bundle is fractional (-0.75), we could call the external boundary of the bundle the fractional Hopfion[35]. Furthermore, all skyrmion bundles can be regarded as skyrmion tubes encircled by a fractional Hopfion, as shown in Fig. 6b, which suggests that skyrmion

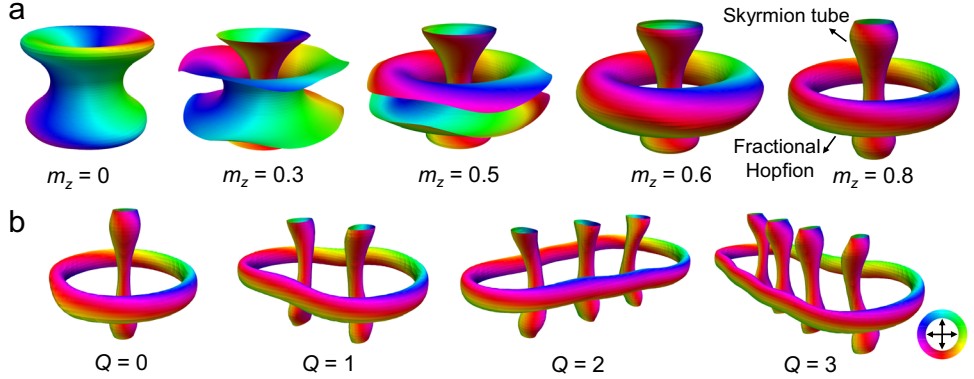

**Fig. 6 | Skyrmion bundles comprise skyrmion tubes encircled by a fractional Hopfion. a** Simulated 3D magnetic configurations of skyrmion bundles. The iso-surfaces correspond to different values of the normalized out-of-plane magnetization component $m_z$. **b** Simulated magnetic iso-surfaces of the $Q = 0, 1, 2,$ and 3 bundles for $m_z = 0.9$. The colorwheel represents the in-plane magnetizations.

bundles could possess the combined topological magnetism of skyrmions and Hopfions.

## Discussion

In summary, we have demonstrated the thermal stability of skyrmion bundles at physical conditions of realistic devices, i.e., room temperature and zero magnetic fields. We propose the creation of skyrmion bundles using a combination of pulsed current and reversed magnetic field without an additional field-cooling procedure. We also demonstrate the room-temperature multi-$Q$ characteristics of skyrmion bundles, including topological quantized annihilation, magnetic phase diagram, and $Q$-related bundle size. Metastable perpendicular helical background magnetization can be stabilized at zero field and low temperatures, which contributes to the stability of skyrmion bundles at zero fields below 320 K. However, the strong thermal fluctuation near Curie temperature led to the only ground horizontal helix domains, resulting in the instability of skyrmion bundles above 320 K. The further increase of thermal stability of skyrmion bundles at room temperature can be expected to be achieved in chiral magnets with higher Curie temperatures. Moreover, we have clarified the topological nature of the boundary spiral of skyrmion bundles as magnetic fractional Hopfions, which suggests the diverse topological magnetism for skyrmion bundles. The stability of skyrmion bundles at both room temperature and zero fields could promote topological spintronic device applications based on the freedom parameter of $Q$.

## Methods

### Sample preparation

Polycrystalline samples of $Co_8Zn_{10}Mn_2$ crystals were synthesized by a high-temperature reaction method. Stoichiometric cobalt (> 99.9%), zinc (> 99.99%), and manganese (> 99.95%) were mixed into a quartz tube and sealed under vacuum, heated to 1273 K for 24 h. Then slowly cooled to 1198 K, and maintained at this temperature for more than 72 h. After that, put the tube in cold water to quench. Finally, a spherical $Co_8Zn_{10}Mn_2$ alloy with metallic luster was obtained.

### Fabrication of $Co_8Zn_{10}Mn_2$ micro-devices

The $Co_8Zn_{10}Mn_2$ micro-devices with a thickness of ~160 nm for TEM observation were fabricated from a polycrystal $Co_8Zn_{10}Mn_2$ alloy by the lift-out method using the focus ion beam (FIB) dual-beam system (Helios NanoLab 600i; FEI). The micro-device employs a silicon-based substrate chip equipped with four Au electrodes. The design incorporates placing the sample at the edge with a suspended thin region, allowing electron beams to pass through the specimen for imaging purposes. The specimen thin film, prepared using FIB

technology, is welded to the chip's edge via $PtC_x$ deposition, and similarly, $PtC_x$ is used to electrically connect the chip electrodes. Ultimately, conductive silver epoxy and gold wires are utilized to join the chip's electrode terminals to those of the specimen holder. A pulse current source is connected through wires to a series of attenuators, then onto the current-carrying specimen holder. Within the specimen holder, internal wiring connects to the specimen mounted above, thereby forming a complete circuit loop. By inserting the specimen holder into the electron microscope, it becomes possible to simultaneously apply pulse currents to the specimen while observing the magnetic structures within the sample under the influence of the current. The detailed procedures can be found in Supplementary Fig. 12.

### TEM measurements

The Lorentz Fresnel imaging was recorded by Lorentz mode at room temperature, and the accelerating voltage of TEM (Talos F200X, FEI) is 200 kV. A perpendicular varying magnetic field is applied by changing the object's current. The current pulses were provided using a voltage source (AVR-E3-B-PN-AC22, Avtech Electrosystems), and the pulse widths were set to 10-200 ns with a frequency of 1 Hz.

### Micromagnetic Simulation

The micromagnetic simulation package JuMag was used to investigate the evolution of skyrmion bundles with magnetic field. The sum of micromagnetic energy $E$ over volume $V_s$ is:

$$E = \int_{V_s} (\varepsilon_e + \varepsilon_a + \varepsilon_D + \varepsilon_z + \varepsilon_d)\mathrm{d}\boldsymbol{r} \tag{5}$$

Here, exchange energy density

$$\varepsilon_e = A(\partial_x \mathbf{m}^2 + \partial_y \mathbf{m}^2 + \partial_z \mathbf{m}^2) \tag{6}$$

anisotropy energy density

$$\varepsilon_a = -K_u(\mathbf{u} \cdot \mathbf{m})^2 \tag{7}$$

Dzyaloshinskii-Moriya interaction energy density

$$\varepsilon_D = D\mathbf{m} \cdot (\nabla \times \mathbf{m}) \tag{8}$$

Zeeman energy density

$$\varepsilon_z = -M_s \mathbf{B} \cdot \mathbf{m} \tag{9}$$

and demagnetization energy density

$$\varepsilon_d = -\frac{1}{2} M_s \mathbf{B}_d \cdot \mathbf{m} \tag{10}$$

The vector **u** is the unit orientation vector, and **m** is the unit vector of magnetization.

We set exchange interaction constant $A = 3.25 \times 10^{-12}$ J m$^{-1}$, saturated magnetization $M_s = 2.78 \times 10^5$ A m$^{-1}$, Dzyaloshinskii-Moriya interaction constant $D = 4.8 \times 10^{-4}$ J m$^{-2}$, and anisotropy constant $K_u = 8.1 \times 10^3$ J m$^{-3}$. The applied external magnetic field **B** is perpendicular to the sample. The thickness of the sample was ~160 nm and the cell size was $2\,nm \times 2\,nm \times 2\,nm$, and the in-plane dimension is $1200 \times 600$ nm. We set the two-dimensional in-plane periodic boundary conditions.

For simulating the current-driven dynamic motions of the skyrmion tube in the continuous of the step edge of nanostructures, a spin-transfer torque term based on the Zhang-Li model was applied with the expression:

$$\varepsilon_{ZL} = \frac{1}{1+\alpha^2} \left\{ (1+\beta\alpha)\mathbf{m} \times [\mathbf{m} \times (\boldsymbol{\mu} \cdot \nabla)\mathbf{m}] + (\beta - \alpha)\mathbf{m} \times (\boldsymbol{\mu} \cdot \nabla)\mathbf{m} \right\} \tag{11}$$

Here

$$\boldsymbol{\mu} = \frac{\mu_B \mu_0}{2e\gamma_0 B_{sat}(1+\beta^2)} P\mathbf{J} \tag{12}$$

and $\mathbf{J}, P, \beta, B_{sat}, \mu_B$ are the current density, the spin current polarization of the chiral magnet, the degree of non-adiabaticity, the Bohr magneton, and the saturation magnetization expressed in Tesla, respectively. We set $\alpha = 0.3$ and $\beta = 0.05$.

## Data availability
The data that support the plots provided in this paper and other finding of this study are available from the corresponding author upon request.

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

## Acknowledgements

This work was supported by the National Key R&D Program of China, Grant No. 2022YFA1403603 (H.D.); the Natural Science Foundation of China, Grants No. 12174396 (J.T.), 12104123 (Y.W.), and 12241406 (H.D.); the National Natural Science Funds for Distinguished Young Scholar, Grant No. 52325105 (H.D.); the Anhui Provincial Natural Science Foundation, Grant No. 2308085Y32 (J.T.); the Natural Science Project of Colleges and Universities in Anhui Province, Grant No. 2022AH030011 (J.T.); the Strategic Priority Research Program of Chinese Academy of Sciences, Grant No. XDB33030100 (H.D.); CAS Project for Young Scientists in Basic Research, Grant No. YSBR-084 (H.D.); Systematic Fundamental Research Program Leveraging Major Scientific and Technological Infrastructure, Chinese Academy of Sciences, Grant No. JZHKYPT-2021-08 (H.D.) ; Anhui Province Excellent Young Teacher Training Project Grant No. YQZD2023067 (Y.W.); and the China Postdoctoral Science Foundation Grant No. 2023M743543 (Y.W.).

## Author contributions

H.D. and J.T. supervised the project. J.T. conceived the idea and designed the experiments. X.X. synthesized $Co_8Zn_{10}Mn_2$ crystals. Y.Z., J.T., and Y.W. fabricated the $Co_8Zn_{10}Mn_2$ microdevices and performed TEM measurements. M.S. and J.T. performed the simulations. J.T., Y.Z., and H.D. wrote the manuscript with input from all authors. M.T., S.W., J.T., H.D., and Y.Z. discussed the results and commented on the manuscript.

## Competing interests

The authors declare no competing interests.
