## [Peer Review File · Nature Communications]

Stable skyrmion bundles at room temperature and zero magnetic field in a chiral magnetEditorial Note: Figure R15b in this Peer Review File is reproduced with permission from Springer Nature. <Zheng, F., et al, Nat. Phys. 18, 863–868 (2022). >

Reviewers' Comments:

Reviewer #1:

Remarks to the Author:

Topological spin textures show high promising for the low energy consuming spintronic devices. The main challenges is that their advances can be realized at room temperature and zero external field. As a novel spin textures, skyrmion bundles has been achieved in chiral magnets at room temperature and zero magnetic field in this work. The authors proposed a method of combining pulsed currents and reversed magnetic fields to obtain skyrmion bundles with different inter-Q values, especially, got a maximum of 24 bundles in β -Mn-type Co₈Zn₁₀Mn₂ chiral magnet. Furthermore, they also demonstrated the field-driven topological quantitated annihilation of high-Q bundles as well as a stable phase diagram of temperature and field confirmed by micromagnetic simulations. Their findings are useful for the development of spintronic devices based on spin textures. I would like to recommend its publication in NC after addressing the following comments.

1. I suggest the authors give a conceptual figures to explain their scenario of creating skyrmion bundles, instead of using experimental results in Figure 1. It will be easier for the audiences' understanding.
2. The data presented in Fig. 2d is extracted from the measured images, which should be points like what shown in Fig. 2b.
3. The data presented in Fig. 2f is not clear. They should explain more in details of how to get this data. And the red solid line is only the "hand-drawing" or theretical calculations?
4. From my understanding that the authors want to compare the measureed results with simulated results in Fig. 3. They should clearly indicated that a and c are the measured results, b and d are the simulated results. From Fig. 3a and 3c, the field ranges used are quite different between the experiments and simulations. Can the authors explain the reasons?
5. Fig. 4c, d, and e are not clear. It would be better to improve them or present them as a separate Fig. 5.
6. Regarding the simulation, the thickness is 160 nm. What is the in-plane dimension? Is the periodic boundary conditions adopted?

Reviewer #2:

Remarks to the Author:

The authors experimentally demonstrate the generation of skyrmion bundles composed of a small number of skyrmions by application of current pulses and gradual variation of magnetic field. If a skyrmion bundle with a small number of skyrmions can be produced at room temperature and zero magnetic field in a controlled manner, they could be useful for spintronics applications as the authors say. The Lorentz-TEM real-space images of skyrmion bundles are beautiful, and the micromagnetic simulations that qualitatively reproduce the experimental results are also commendable. However, after carefully reading the manuscript, we believe that this paper is not worthy of publication in Nature Communications for the following reasons.

1. _The authors aim to generate skyrmion bundles at room temperature and zero field, but the actual experimental results are far from such a goal. The authors first observed evolutional change of conical phase into skyrmions and skyrmion lattice by successive application of current pulses to the conical phase. However, as seen in Fig. 1, what is produced by the current pulses is not a skyrmion bundle composed of a few skyrmions, but rather domains of distorted skyrmion lattices, the smallest of which is composed of 100 to 300 skyrmions.
2. The authors then generate skyrmion bundles by starting from the skyrmion lattice phase and gradually varying the magnetic field. However, these skyrmion bundles are not manifested at zero magnetic field, but when a strong magnetic field is applied. Since skyrmions are topological defects, it

is a simple expectation that quantized changes in topological charges can be observed if the magnetic field is gradually increased during the transition to the ferromagnetic state. The experimental results are totally opposite to the authors' original goal of generating skyrmion bundles at zero magnetic field. These skyrmion bundles under a relatively strong magnetic field should quickly turn into ordinary skyrmion lattice phase if the magnetic field is reduced to zero.

3. Although it is claimed that the skyrmion bundles have potential to be applied to topological spintronics devices, there is no concrete discussion on the possible directions of their technical applications and the expected device functionalities.

4. The generation of domains of skyrmion crystals by stimulation with current pulses by taking advantage of the first-order transition nature at the phase boundary between the conical magnetic phase and the skyrmion lattice phase and the quantized annihilation of skyrmions with increasing magnetic fields are results that are within the range of naive expectations and do not seem to contain any particularly important or surprising physics that deserve publication in Nature Communications.

Overall, I cannot recommend publication of this manuscript in Nature Communications.

Reviewer #3:

Remarks to the Author:

The authors present an electron microscopy study of skyrmion bundles found in lamellae of the chiral magnet Co₈Zn₁₀Mn₂. The work is a continuation of their previous study, which featured the initial discovery of skyrmion bundles in FeGe, published in Nat. Nanotechnol. in 2021. Overall, the present work makes two advances compared to the previous study: firstly through the observation of these interesting 3D magnetic states at room temperature, and also through the exploration of their stability in field and temperature.

I found the data convincing, particularly the agreement between the experimental and simulation data, and the manuscript reasonably well written, although I will point out a few minor mistakes below, and the English should be checked through again.

Overall, my main concern is that the work is quite similar to the authors' previous Nat. Nanotechnol. publication. Can the authors better motivate the distinction between this and their previous work, other than the higher temperature? For example, the authors relegate their observation of the bound state of an up and down skyrmion to supplementary, but has anyone observed this state before? That seems like an interesting finding on its own. I detail more comments below.

1) At the end of the introduction, the authors state they have "reported the unambiguous experimental realization of a type of 3D multi-Q skyrmionic configurations". To be clear, I agree with the authors. However, to me the lack of any corresponding 3D image/schematic from the main text was surprising. My suggestion would be to include some kind of 3D image in Fig. 1, and perhaps also a SEM picture of their device structure. I think this is well-motivated by the presence of the paragraph between lines 146-159, where the authors discuss the 3D texture while heavily referencing supplementary Fig. S3. Why not move this to the main text?

2) The methods section is extremely limited. Could the authors include more details of the fabrication and experimental procedure? What kind of substrate was their device structure mounted to, and how was it contacted? Readers should be able to fully replicate the experiment from the descriptions in the methods section.

Some more minor comments:

3) Across the manuscript, the authors refer to "electro-magneto" or "magneto-electro" properties. I would suggest unifying these and writing "electromagnetic".

4) Line 27: I think the authors mean "exclusive"  "elusive"?

5) Throughout the manuscript, the authors write "inter-Q". I think they probably mean "integer Q"?

6) Line 41: I think the authors mean "underpinning""depinning"?

7) Perhaps the authors might very briefly introduce the $\text{Co}_8\text{Zn}_{10}\text{Mn}_2$ material (at least state it is a bulk chiral magnet with DMI-stabilised skyrmions), around line 84.

Note that all page numbers and references without special instructions refer to the newly revised manuscript attached with this response and not to the original version submitted. The added contents are marked in red in the revised manuscript.

Response to Reviewer #1:

Reviewer #1: Topological spin textures show high promising for the low energy consuming spintronic devices. The main challenges is that their advances can be realized at room temperature and zero external field. As a novel spin textures, skyrmion bundles has been achieved in chiral magnets at room temperature and zero magnetic field in this work. The authors proposed a method of combining pulsed currents and reversed magnetic fields to obtain skyrmion bundles with different inter-Q values, especially, got a maximum of 24 bundles in β -Mn-type $\text{Co}_8\text{Zn}_{10}\text{Mn}_2$ chiral magnet. Furthermore, they also demonstrated the field-driven topological quantitated annihilation of high-Q bundles as well as a stable phase diagram of temperature and field confirmed by micromagnetic simulations. Their findings are useful for the development of spintronic devices based on spin textures. I would like to recommend its publication in NC after addressing the following comments.

Response: We thank you for your positive comments on our manuscript and appreciate the reviewer for thinking that our manuscript is *useful for the development of spintronic devices based on spin textures*. Below please find our response to each helpful comment.

Comment 1: I suggest the authors give a conceptual figures to explain their scenario of creating skyrmion bundles, instead of using experimental results in Figure 1. It will be easier for the audiences' understanding.

Response: We thank you for the insightful suggestion. In our revised manuscript, we have incorporated a figure that illustrates the process of generating skyrmion bundles at room temperature and zero field, as shown in Fig. R1. The experimental sequence commences with the induction of conical-to-skyrmion transformations through pulsed currents applied under negative magnetic fields. Subsequently, the formed skyrmion lattice is observed to remain stable at zero field and further evolves into skyrmion bundles upon application of positive magnetic fields. Notably, these skyrmion bundles continue to exist even when the magnetic field is reduced back to zero. It is worth emphasizing that all these experimental procedures can be performed at room temperature.

Fig. R1 (*i.e.*, Fig. 1a) Scenario of creating skyrmion bundles. The images below each display corresponding simulated Fresnel images.

Comment 2: The data presented in Fig. 2d is extracted from the measured images, which should be points like what shown in Fig. 2b.

Response: We thank the reviewer for careful reading, and we have presented the data shown in Figure as point form, as shown in Fig. R2.

Fig. R2 (*i.e.*, Fig. 3d) Line profile of magnetic phase shift.

Comment 3: The data presented in Fig. 2f is not clear. They should explain more in details of how to get this data. And the red solid line is only the “hand-drawing” or theoretical calculations?

Response: We have added detail to show how to get the presented data in Fig. 2f. We first determined the boundary of skyrmion bundles from the line profile of in-plane magnetization magnitude M_{xy} . We define the location of boundary as the maximum M_{xy} within the closure helical domain (marked as A and B), as shown in Fig. R3. Then we calculate the full area inside of the boundary (*i.e.* the red dashed line).

Fig. R3 (i.e., Supplemental Fig. S3) The flowchart of skyrmion bundles area determination

See lines 180-185: “The internal skyrmions of the bundles always tightly bind to each other and have a comparable size as that of an isolate skyrmion, leading to a linear increased area S as a function of Q . We were able to identify the boundaries of skyrmion bundles and subsequently calculated the area within these determined boundaries (Supplemental Fig. S3), which represents the area occupied by skyrmion bundles.”

Fig. R4 (i.e., Fig. 3f) Dependence between the area S of different skyrmion bundles and topological charge Q . The fitting formula is $S = (Q + 1)(S_{N-skr} + S_{spiral})$.

The red solid line indicates the result of a linear fit performed on the experimental data points. We have specified the representation of dots and red solid line in the caption of Fig. 3 in the main text.

Comment 4: From my understanding that the authors want to compare the measured results with simulated results in Fig. 3. They should clearly indicated that a and c are the measured results, b and d are the simulated results. From Fig. 3a and 3c, the field ranges used are quite different between the experiments and simulations. Can the authors explain the reasons?

Response: We thank you for careful reading, we have distinctly marked the experimental results and simulation results in Fig. 4 in the revised manuscript, aiming to provide readers with a more intuitive differentiation. See the caption of Fig. 4 in Lines 230-234: “**Fig. 4 | Field-driven quantized topological annihilations. a**, Field-driven magnetic evolutions from a skyrmion bundle with $Q = 8$ at 295 K. **b**, Corresponding

simulated averaged in-plane magnetization mapping during the field-driven magnetic evolution. **c**, Topological charge Q as a function of magnetic field B . **d**, Corresponding simulated topological charge Q as a function of magnetic field B .”

We set the simulation parameters based on a previous work with same targeting material [Nat. Commun. 2022, 13 (1), 7], which adopt material parameters in low temperatures. The disparity in the magnetic field ranges between experimental results and simulations can be attributed to the fact that simulations operate under zero-temperature conditions. As our experiments were carried out over a range of temperatures, the experimental parameters are inherently temperature-sensitive, while the simulations were executed at zero-temperature. However, it's noteworthy that as the experimental temperature decreases, the field range required for the topological transformations of skyrmion bundles converges towards the simulated field range. Despite the variation in field ranges between experimental and simulation scenarios, the magnetic configurations and associated topological transitions remain consistent.

Comment 5: Fig. 4c, d, and e are not clear. It would be better to improve them or present them as a separate Fig. 5.

Response: We thank the reviewer for careful reading, we have converted these three images into vector graphics for clearer presentation in the revised manuscript, as shown in Fig. R5.

Fig. R5 (*i.e.*, Fig. 5d-f) Magnetic phase diagrams of the skyrmion bundles ($Q = 0, 1,$ and 2) as a function of temperature and magnetic field. The data points in the diagram are the critical points of the two-state transition. FM and PM represent ferromagnetic and paramagnetic states, respectively.

Comment 6: Regarding the simulation, the thickness is 160 nm. What is the in-plane dimension? Is the periodic boundary conditions adopted?

Response: Regarding the simulation, the in-plane dimension is $1200 \text{ nm} \times 600 \text{ nm}$, employing periodic boundary conditions across both dimensions. For enhanced clarity, we have included a more detailed description about simulations, see Lines 485-486: “*The in-plane dimension is $1200 \times 600 \text{ nm}$. We set the two-dimensional in-plane periodic boundary conditions.*”

Response to Reviewer #2:

Reviewer #2: The authors experimentally demonstrate the generation of skyrmion bundles composed of a small number of skyrmions by application of current pulses and gradual variation of magnetic field. If a skyrmion bundle with a small number of skyrmions can be produced at room temperature and zero magnetic field in a controlled manner, they could be useful for spintronics applications as the authors say. The Lorentz-TEM real-space images of skyrmion bundles are beautiful, and the micromagnetic simulations that qualitatively reproduce the experimental results are also commendable. However, after carefully reading the manuscript, we believe that this paper is not worthy of publication in *Nature Communications* for the following reasons.

Response: We express our profound gratitude for the meticulous effort you dedicated to reviewing our manuscript. We have meticulously addressed your constructive feedback and thoroughly revised the manuscript accordingly. We earnestly request you to reconsider our submission for publication in *Nature Communications*. Hereafter, we present our detailed responses to each of your invaluable comments.

Comment 1: The authors aim to generate skyrmion bundles at room temperature and zero field, but the actual experimental results are far from such a goal. The authors first observed evolutionary change of conical phase into skyrmions and skyrmion lattice by successive application of current pulses to the conical phase. However, as seen in Fig. 1, what is produced by the current pulses is not a skyrmion bundle composed of a few skyrmions, but rather domains of distorted skyrmion lattices, the smallest of which is composed of 100 to 300 skyrmions.

The authors then generate skyrmion bundles by starting from the skyrmion lattice phase and gradually varying the magnetic field. However, these skyrmion bundles are not manifested at zero magnetic field, but when a strong magnetic field is applied. Since skyrmions are topological defects, it is a simple expectation that quantized changes in topological charges can be observed if the magnetic field is gradually increased during the transition to the ferromagnetic state. The experimental results are totally opposite to the authors' original goal of generating skyrmion bundles at zero magnetic field. These skyrmion bundles under a relatively strong magnetic field should quickly turn into ordinary skyrmion lattice phase if the magnetic field is reduced to zero.

Response: We thank you for your careful consideration. We agree with you for the creation of skyrmion bundles using two steps, *i.e.* pulsed currents and reversed magnetic fields. However, we kindly suggest that skyrmion bundles do not turn into ordinary skyrmion lattice when the magnetic field is reduced to zero. As shown in Fig. R6 and R7, both simulation and experimental evidence confirm that skyrmion bundles with varying Q can persistently retain their structure when the magnetic field intensity is decreased to zero.

Fig. R6 (*i.e.*, Supplemental Fig. S6a-b) Simulated skyrmion bundles under zero field.

Fig. R7 (*i.e.*, Supplemental Fig. S6c-d) Experimental skyrmion bundles under zero field.

The stability of skyrmion bundles under zero field stems from a unique configuration known as stacked spirals, which exhibit no Fresnel contrasts, as shown in Fig. R8. Here, we propose the achievement of stable isolate topological spin textures under zero field in the magnetization conical background. In our revised manuscript, prior to delving into the stability of skyrmion bundles under zero field, we initially show the stability of an isolate skyrmion under zero field, as shown in Fig. R9.

Fig. R8 (*i.e.* Supplemental Fig. S7) Zero-field stability of stacked spiral. a-c, Fresnel images from FM, to conical, and finally to stacked spiral state as the magnetic field decreases continuously. Defocus distance, $-1200 \mu\text{m}$. d-f, Corresponding simulated schematic diagrams of the three-dimensional structures from FM at 350 mT, to conical at 69 mT, and finally to stacked spiral state at zero magnetic field. The images below each display the corresponding simulated Fresnel images.

Fig. R9 (*i.e.* Supplemental Fig. S5) stability of an isolate skyrmion under zero magnetic field. FM and PM represent ferromagnetic and paramagnetic states, respectively.

Upon reducing the magnetic field to zero, the background magnetization transforms into a spiral configuration, which serves as a supportive foundation for the stability of isolated skyrmion bundles. While helical magnetization may indeed exhibit a lower energy state under zero field, we always prefer to obtain stacked spiral background magnetization other than helical phase following the energy path as shown in Fig. R8. Zero-field magnetic skyrmion bundles can be obtained in a broad temperature range far

below the Curie temperature. However, as the temperature approaches the vicinity of the Curie point, helical structures start to dominate, as shown in Fig. R10.

Fig. R10 (*i.e.* Supplemental Fig. S8) Instability of skyrmion bundles and stacked spirals above 320 K. The process of skyrmion bundle with $Q = 0$ rising temperature from 200 K to 320 K under zero-field.

Ultimately, by decreasing field to zero from bundles at various temperatures, we obtain the stability diagram of skyrmion bundles within the low-field regime, as shown in Fig. R11. These results well confirm the stability of skyrmion bundles at zero magnetic fields and room temperature.

Fig. R11 (*i.e.* Fig. 5d-f) Magnetic phase diagrams of the skyrmion bundles ($Q = 0, 1, 2$) as a function of temperature and magnetic field. The data points in the diagram are the critical points of the two-state transition. FM and PM represent ferromagnetic and paramagnetic states, respectively.

Comment 3: Although it is claimed that the skyrmion bundles have potential to be applied to topological spintronics devices, there is no concrete discussion on the possible directions of their technical applications and the expected device functionalities.

Response: We greatly appreciate your meticulous reading. In response, we have augmented concrete discussion on potential technological applications and the anticipated functionalities of devices exploiting skyrmion bundles. Skyrmion bundles provide numerous new information carriers with distinct topology-related magnetism and dynamics, which could promote topological the development of spintronic devices, such as binary racetrack memories using any two types of skyrmion bundles, multiple-bits ASCII information encoding, and interconnect devices.

See Lines 48-53: “Skyrmion bundles, holding the advantages of diverse Q -related electro-magneto properties and morphologies, have greatly enriched the family of topological magnetic solitons and *shown potential application in extended topological spintronic devices, such as binary racetrack memories using any two types of skyrmion bundles, multiple-bits ASCII information encoding, and interconnect devices*²³⁻²⁸.”

Comment 4: The generation of domains of skyrmion crystals by stimulation with current pulses by taking advantage of the first-order transition nature at the phase boundary between the conical magnetic phase and the skyrmion lattice phase and the quantized annihilation of skyrmions with increasing magnetic fields are results that are within the range of naive expectations and do not seem to contain any particularly important or surprising physics that deserve publication in Nature Communications.

Response: We believe the advancement of our manuscript contains the following aspects: 1st, the technical advancement of creating skyrmion bundles in our manuscript is the employment of pulsed current replacing the traditional complex field-cooling process [*Nat. Nanotechnol.* **2021**, 16:1086. *Adv. Mater.* **2021**, 33, 2004110] to create skyrmions at negative magnetic fields; 2nd, the realization of chiral skyrmion bundles at room temperatures; 3rd, from the view of important physics, we demonstrate that isolated skyrmion bundles can be stabilized under zero magnetic field in the spiral background magnetizations, which is different from the application of geometrical confinement effects, see discussion in Lines 236-242: “*The stability of zero field topological spin textures has been typically realized in geometrically confined nanostructures or a compact lattice*^{45,48-50}. *It is important to further achieve isolated topological solitons in free geometries. Here, we show that isolated zero-field skyrmion bundles can be created in the stacked spiral magnetization without the application of geometrical confinement effects.*”

Moreover, upon meticulous examination of spin configurations in our simulations, we conclude that the new topological physical nature of skyrmion bundles, *i.e.* skyrmion tubes encircled by fractional Hopfions. Consequently, it is open to exploring emergent topological magnetism of skyrmion bundles including skyrmions and Hopfions.

Fig. R12 (*i.e.* Fig. 6) Skyrmion bundles comprise skyrmion tubes encircled by a fractional Hopfion. a. Simulated 3D magnetic configurations of skyrmion bundles. The iso-surfaces correspond to different values of the normalized out-of-plane magnetization component m_z . b. Simulated magnetic iso-surfaces of the $Q = 0, 1, 2,$ and 3 bundles for $m_z = 0.9$. The color represents the in-plane magnetization m_{xy} .

Following your suggestions “*If a skyrmion bundle with a small number of skyrmions can be produced at room temperature and zero magnetic field in a controlled manner, they could be useful for spintronics applications as the authors say*”, we have illustrated a scenario of creating skyrmion bundles at room temperature and zero fields, as shown in Fig. R1. Using such a procedure, we could create skyrmion bundles with different Q at room temperature and zero fields, as shown in Fig. R13.

Fig. R13 (*i.e.* Fig. 5c) Experimental skyrmion bundles at room temperature and zero fields.

In summary, our manuscript fulfills a stable diagram of the skyrmion bundle as a function of field and temperature as well as the topological magnetism of bundles. We sincerely request you to reconsider our manuscript’s publication in *Nature Communications*.

Response to Reviewer #3:

Reviewer #3: The authors present an electron microscopy study of skyrmion bundles found in lamellae of the chiral magnet $\text{Co}_8\text{Zn}_{10}\text{Mn}_2$. The work is a continuation of their previous study, which featured the initial discovery of skyrmion bundles in FeGe, published in Nat. Nanotechnol. in 2021. Overall, the present work makes two advances compared to the previous study: firstly through the observation of these interesting 3D magnetic states at room temperature, and also through the exploration of their stability in field and temperature.

I found the data convincing, particularly the agreement between the experimental and simulation data, and the manuscript reasonably well written, although I will point out a few minor mistakes below, and the English should be checked through again.

Response: We are deeply grateful for the considerable time and effort you invested in reviewing our manuscript. Your meticulous attention to detail has been instrumental in identifying several minor errors, which we have since rectified through a thorough language review.

Overall, my main concern is that the work is quite similar to the authors' previous Nat. Nanotechnol publication. Can the authors better motivate the distinction between this and their previous work, other than the higher temperature?

Response: We thank you for your concern. Beyond the high-temperature stability of skyrmion bundles, we believe the advancement of our manuscript includes the following aspects: First, we provide the stability of skyrmion bundles under zero magnetic field using the application of stacked spiral background magnetizations, thus we complete the full temperature-field diagram for the stability of skyrmion bundles; Second, unlike our previous method, we introduce a new step-1 procedure for generating skyrmion bundles that employ pulsed currents instead of field-cooling from elevated temperatures to produce skyrmions at negative fields; Third, we have also added discussion between skyrmion bundles and Hopfions, and our further analysis indicate that skyrmion bundles can be considered as the combination of fractional Hopfions and skyrmion tubes.

In our revised manuscript, we have provided the details about the role of stacked spiral magnetization in stabilizing isolate skyrmion bundles as well skyrmions at zero magnetic fields, see Lines 236-242: *“The stability of zero field topological spin textures has been typically realized in geometrically confined nanostructures or a compact lattice^{45,48-50}. It is important to further achieve isolated topological solitons in free geometries. Here, we show that isolated zero-field skyrmion bundles can be created in the stacked spiral magnetization without the application of geometrical confinement effects.”*

We have added a new section in discussing the fractional Hopfion of skyrmion bundles. See Lines 283-307: “**Fractional Hopfion rings in skyrmion bundles.** *Magnetic Hopfions are topologically stable, three-dimensional magnetic structures that exhibit unique and fascinating properties⁵¹⁻⁵⁴. These localized spin configurations arise in magnetic materials with nontrivial symmetry, such as chiral magnets or frustrated systems. Unlike conventional magnetic domain walls or skyrmions that have a one-dimensional or two-dimensional structure, respectively, Hopfions possess a complex three-dimensional magnetic texture resembling a toroidal knot. Due to their stability and intriguing characteristics, Hopfions have attracted significant attention in the field of spintronics and magnetic storage devices. Understanding the fundamental properties and dynamics of these magnetic structures holds great promise for developing novel technologies in the future. The topological index of Hopfions can be mathematically represented by the equation $Q_H = -\frac{1}{(8\pi)^2} \int \mathbf{F} \cdot \mathbf{A} dr$, where \mathbf{A} represents the vector potential of $\mathbf{F} = \nabla \times \mathbf{A}$. The vector \mathbf{F} can be expressed using the local magnetization direction, represented by the unit vector \mathbf{n} , and the Levi-Civita permutation symbol ε . Specifically, the components of F_i can be defined as $F_i = \varepsilon_{ijk} \mathbf{n} \cdot (\partial_j \mathbf{n} \times \partial_k \mathbf{n})$.*

We then explore the relationship between skyrmion bundles and magnetic Hopfions. Taking $Q = 0$ bundles as an example, we show different iso-surfaces for different m_z values, as shown in Fig. 6a. For iso-surfaces of $m_z > 0.6$, the in-plane magnetizations m_{xy} reveal the same characteristics as the magnetic Hopfion. As the Hopfion charge Q_H of the $Q = 0$ bundle is fractional (~ 0.75), we could call the external boundary of the bundle the fractional Hopfion⁵³. Furthermore, all skyrmion bundles can be regarded as skyrmion tubes encircled by a fractional Hopfion, as shown in Fig. 6b, which suggests that skyrmion bundles could possess the combined topological magnetism of skyrmions and Hopfions.”

Fig. R14 (i.e. Fig. 6) Skyrmion bundles comprise skyrmion tubes encircled by a fractional Hopfion. a. Simulated 3D magnetic configurations of skyrmion bundles. The iso-surfaces correspond to different values of the normalized out-of-plane magnetization component m_z . b.

Simulated magnetic iso-surfaces of the $Q = 0, 1, 2,$ and 3 bundles for $m_z = 0.9$. The color represents the in-plane magnetization m_{xy} .

For example, the authors relegate their observation of the bound state of an up and down skyrmion to supplementary, but has anyone observed this state before? That seems like an interesting finding on its own.

Response: We present the supplemental bound state of an up-and-down skyrmion to show that diverse topological magnetic states can be obtained using the combination of pulsed currents and reversed magnetic fields. A previous study [*Nat. Phys.* **2022**, *18*, 863-868] has discussed the skyrmion-antiskyrmion pair. However, it seems that our experimental Fresnel contrast looks very similar to the ideal bounded skyrmion-antiskyrmion pair predicted in simulations. The other achievement is the room temperature instead of 95 K observation of the bounded skyrmion-antiskyrmion pair.

We have added a discussion about the bounded skyrmion-antiskyrmion pair in lines 188-191: *“It should be noted that the combination of pulsed current and reversed magnetic fields can be a promising technique in achieving other fascinating topological solitons, such as the closely bounded skyrmion-antiskyrmion pair (Supplemental Fig. S4) [44], at room temperature.”*

Fig. R15 a. A skyrmion-antiskyrmion pair induced by the in-plane current at room temperature and corresponding simulated Lorentz TEM image. b. Experimental Lorentz TEM image of a skyrmion-antiskyrmion pair at 95 K in a thin FeGe sample and corresponding simulated Lorentz TEM image.

I detail more comments below.

Response: Below please find our response to each helpful comments.

Comment 1: At the end of the introduction, the authors state they have “reported the unambiguous experimental realization of a type of 3D multi-Q skyrmionic configurations”. To be clear, I agree with the authors. However, to me the lack of any corresponding 3D image/schematic from the main text was surprising. My suggestion would be to include some kind of 3D image in Fig. 1, and perhaps also a SEM picture of their device structure. I think this is well-motivated by the presence of the paragraph between lines 146-159, where the authors discuss the 3D texture while heavily referencing supplementary Fig. S3. Why not move this to the main text?

Response: We thank the reviewer for careful reading, we have moved the 3D schematic of skyrmion bundles from the supplementary Fig. S3 into the main text, and we have also included the image of device structure in the revised manuscript, as shown R16.

Figure R16 (i.e. Fig. 2) **a.** The image of device structure obtained from scanning electron microscopy imaging. **b-d,** A sequence of Fresnel images of the skyrmion creation process after applying numbers of the current pulse, corresponding to conical, skyrmions, and skyrmion lattice, respectively. The red rectangular dashed lines are the thin area on both sides of the sample in **b.** Inset: in-plane magnetization mapping of skyrmion lattice within a hexagonal dashed line by Transport of Intensity Equation (TIE).

Comment 2: The methods section is extremely limited. Could the authors include more details of the fabrication and experimental procedure? What kind of substrate was their device structure mounted to, and how was it contacted? Readers should be able to fully replicate the experiment from the descriptions in the methods section.

Response: We thank the reviewer for careful reading, we have added more details of the fabrication and experimental procedure in methods section.

See Lines 453-468: “The $\text{Co}_8\text{Zn}_{10}\text{Mn}_2$ micro-devices with a thickness of ~ 160 nm for TEM observation were fabricated from a polycrystal $\text{Co}_8\text{Zn}_{10}\text{Mn}_2$ alloy by the lift-out method using the focus ion beam (FIB) dual-beam system (Helios NanoLab 600i; FEI).

The micro-device employs a silicon-based substrate chip equipped with four Au electrodes. The design incorporates placing the sample at the edge with a suspended thin region, allowing electron beams to pass through the specimen for imaging purposes. The specimen thin film, prepared using FIB technology, is welded to the chip's edge via PtC_x deposition, and similarly, PtC_x is used to electrically connect the chip electrodes. Ultimately, conductive silver epoxy and gold wires are utilized to join the chip's electrode terminals to those of the specimen holder. A pulse current source

is connected through wires to a series of attenuators, then onto the current-carrying specimen holder. Within the specimen holder, internal wiring connects to the specimen mounted above, thereby forming a complete circuit loop. By inserting the specimen holder into the electron microscope, it becomes possible to simultaneously apply pulse currents to the specimen while observing the magnetic structures within the sample under the influence of the current. The detailed procedures can be found in Supplementary Fig. S11.”

Figure R17 (i.e. Supplemental Fig. S11) The structure of a chip equipped with four Au electrodes and its connection to the sample holder. **a**, The sample holder loaded with a chip and sample. **b**, Magnified area of the red dashed line in **a**. **c**, Magnified area of the red dashed line in **b**, and displays the structure of a chip. **d**, Magnified area of the red dashed line in **c**, and illustrates the placement of the sample and the method of connection to the Au electrodes.

Comment 3: Across the manuscript, the authors refer to “electro-magneto” or “magneto-electro” properties. I would suggest unifying these and writing “electromagnetic”.

Response: We thank the reviewer for careful reading, and we have unified these terms and written them as “electromagnetic” in revised manuscript.

See lines 21-22: “Topological spin textures are characterized by topological magnetic charges Q , which govern their fascinating *electromagnetic* properties.” and lines 42-44: “Despite the significance of Q in determining *electromagnetic* properties of topological spin textures¹¹⁻¹⁹, traditional skyrmions are constrained to possess a fixed value of $|Q| = 1$.”

Comment 4: Line 27: I think the authors mean “exclusive”  “elusive”?

Response: We thank the reviewer for careful reading, and we have corrected “exclusive” to “elusive” in revised manuscript.

See lines 24-27: “However, the realization of stable skyrmion bundles in chiral magnets at room temperature and zero magnetic field, which is the prerequisite for realistic device applications, has remained *elusive*.”

Comment 5: Throughout the manuscript, the authors write “inter-Q”. I think they probably mean “integer Q”?

Response: We thank the reviewer for careful reading, and we have corrected “inter-Q” to “integer Q ” in revised manuscript.

See lines 28-30: “we experimentally achieved skyrmion bundles with different *integer* Q values, reaching a maximum of 24 at above room temperature and zero magnetic field in a β -Mn-type $\text{Co}_8\text{Zn}_{10}\text{Mn}_2$ chiral magnet.” and lines 63-64: “In this work, we successfully observed skyrmion bundles with varying *integer* Q values, including a remarkable maximum of 24.”

Comment 6: Line 41: I think the authors mean “underpinning””depinning”?

Response: We thank the reviewer for careful reading, and we have corrected “underpinning” to “depinning” in revised manuscript.

See lines 38-41: “Topological magnetic charge Q plays a crucial role in determining various topology-related properties of skyrmions, including skyrmion Hall effects^{4, 5}, topological Hall effects⁶, ultrasmall *depinning* current⁷, particle-like physics⁸, and electric transport properties^{9, 10}.”

Comment 7: Perhaps the authors might very briefly introduce the $\text{Co}_8\text{Zn}_{10}\text{Mn}_2$ material (at least state it is a bulk chiral magnet with DMI-stabilised skyrmions), around line 84.

Response: We thank the reviewer for careful reading, and we have added a brief description of $\text{Co}_8\text{Zn}_{10}\text{Mn}_2$ in revised manuscript.

See Lines 88-91: “The micro-device utilized in the experiment comprises two Pt electrodes and a ~ 160 nm thick lamella with two narrow regions of ~ 80 nm thickness on both sides, fabricated from $\text{Co}_8\text{Zn}_{10}\text{Mn}_2$ *that is a bulk chiral magnet with DMI-stabilized skyrmions*.”

Reviewers' Comments:

Reviewer #1:

Remarks to the Author:

I would like to recommend its publication in NC since the authors have addressed all my comments properly. And I believe that the results presented in this work can enrich the research area of skyrmionics and spintronics.

Reviewer #2:

Remarks to the Author:

I have carefully read replies from the authors to three reviewers including me and the revised manuscript. As a result, I found that the authors have sincerely addressed all the comments, questions and criticisms raised by the three reviewers.

In my previous report, I criticized that the demonstrated skyrmion-bundle generation is far from the original purpose of this experiment which aimed to create stable skyrmion bundles at zero magnetic field. Actually, when I read the original manuscript, I thought that the generated skyrmion bundles survive only under application of magnetic field after reversing the magnetic field. However, after reading the authors reply to my criticism, I realized that I misunderstood the complicated procedure of the skyrmion-bundle generation and that the skyrmion bundles survive even at zero magnetic field.

I think that this aspect was not clearly presented in the previous manuscript. The authors added new conceptual figures (Fig.1) to explain their scenario of creating skyrmion bundles as well as detailed explanations of the procedure in the manuscript. As a result, the advancement of their proposal is now clearly presented.

As for my questions about possible technical application of skyrmion bundles and new important physics contained in the present work, the authors provided convincing answers and revised the manuscript and the supplemental information by adding new argument, figures and references. I found them of satisfactory level. I agree that the experimental demonstration of skyrmion bundles stable at zero field and room temperature without any geometrical confinement is very important for the research on the topological magnetism. Furthermore, the concept of fractional hopfions in the skyrmion bundles explained with a newly added figure (Fig.6) is interesting and important for the fundamental science.

After the entire revisions by the authors, I think that the advancement of the proposed method and the important new physics lying in the experimental results are clearly presented in this paper. After careful consideration, I have changed my previous mind against publication of this work in Nature Communications and now recommend its publication.

Reviewer #3:

Remarks to the Author:

The authors have answered most of my original questions. They have improved the manuscript in several ways, for example by including a new Fig. 1 to better explain the 3D structure of the skyrmion bundles, and by including new simulations about possible fractional hopfion states.

However, for me, this still does not help sufficiently distinguish the work from their previous report on FeGe. From what I can see, the other referees agree that the quality of the data and experimental practices is not in doubt – they are both excellent. Therefore, the question comes down to the novelty of the main conclusions relative to published works.

The authors have three main conclusions, which I evaluate below:

1) The observation of skyrmion bundles at room temperature and zero applied magnetic field: This is well-evidenced throughout the paper, and is the best claim of novelty. I think the comments of referee #2 miss the mark here: yes, the skyrmion bundles are not generated by a current pulse at zero field directly, but following the field manipulation procedure, the skyrmion bundles do survive at zero field at room temperature.

2) The ability to nucleate skyrmion bundles without field-cooling, instead by applying a current pulse: The authors create the precursor states for the skyrmion bundles by applying a current pulse. As the authors themselves admit, this is in reality a Joule heating effect (indeed, both of the references the authors' cite about this point, references 29 and 36, are about Joule heating nucleation of skyrmions). Thus, the authors actually are field-cooling their sample. They are just doing so with a local Joule heating effect. And importantly, they cannot nucleate the skyrmion bundles directly. I think their nucleation method still necessitates the Joule heating.

3) The mapping of a magnetic phase diagram for the skyrmion bundle stability: This claim is clearly demonstrated, as evidenced in Fig. 5. However, similarly to referee #2, I don't know if we should be surprised by this data. It is in keeping with my expectation that the skyrmions would only be stable for small applied fields, since the internal skyrmions are inherently energetically unfavourable in the oppositely applied magnetic field.

The authors have added an additional section concerning fractional hopfions and hopfions (presumably in reaction to two recent publications [X. Z. Yu et al. *Advanced Materials* 35, 2210646 (2023), F. Zheng, et al. *Nature* 623, 718-723 (2023)]). I agree with the authors that their skyrmion bundles can probably be related to both of these topological states, and this is an interesting point of discussion (actually, maybe the skyrmion bundles are quite similar to the hopfion rings claimed in the latter reference). However, at the moment this section feels very disjointed by the way which the authors have squeezed it into the end of the manuscript, and no reference to this part is made in the abstract, introduction or conclusions. I'm also not convinced it presents a significant advance in the understanding of these 3D hopfion-like states, in the present form.

Following this evaluation, I am still not convinced that the work warrants publication in *Nature Communications*. The authors present multiple incremental advances on the topic of these interesting 3D spin textures, but I believe these advances are within the expectations of the work they presented in their previous *Nature Nanotechnology* on skyrmion bundles in FeGe, and the two recent hopfion publications. At the moment, I cannot recommend publication.

I think the authors are in the position to answer several key questions about these 3D spin textures: namely, the detailed 3D structure and topology of the magnetic states, and what is/isn't distinguishing these skyrmion bundles from hopfion states. However, for the moment these questions are not directly addressed in the present manuscript, and it is quite a different topic to what the authors have chosen to focus on (namely, zero field, room temperature skyrmion bundles).

I give a few more detailed comments below:

1) The new Figure 1 is much cleaner, and now I understand a little better the 3D structure of the skyrmion bundles. It's very interesting that the top and bottom layers have a very different topological configuration compared to the middle layers, despite having the same $Q=2$ topological charge. At the moment, the description of Fig. 1 in the main text is still lacking, and I was still confused about what the surface chiral state was – it looks like two antiskyrmion-like textures are the objects contributing to the topological charge. I found later on in the manuscript that this description is indeed correct, when the authors make the same statement. The authors might consider moving this description to

the section describing the Fig. 1.

2) As mentioned, the configuration of the surface and middle states are very different, although the topological charge remains $Q=2$. I think this could be very interesting, because it means that there should be some complex modulation along the vertical axis. Maybe even Bloch points or magnetic monopoles somewhere in the 3D structure, in order to convert between these magnetic structures along the vertical axis. I am interested if this is true, or whether it really is a smooth deformation between these two very different configurations. This reminds me of recent works on the quadrupole Bloch point configuration in antiskyrmions – one of which was authored by some of the present authors [F. Yasin et al. *Advanced Materials* 2311737 (2024), J. Tang et al. *Science Bulletin* 68, 2919-2923]. Could there perhaps be similar interesting 3D structure, if the authors take a closer look into the details of the simulated skyrmion bundles. How does the magnetic configuration change so much through the vertical thickness? Or is it really a smooth change between two configurations of $Q=2$ topology?

3) In several places, the authors discuss the zero-field state of their sample as a “stacked spin spiral”. However, this phrase was coined in another context [F. N. Rybakov et al. *New Journal of Physics* 18, 045002 (2016)], and I don’t believe this is what the authors are claiming to have seen in their work. I think in the present work, it is just a typical magnetic helix which is pinned in the out-of-plane direction. The present state is lacking the long-wavelength modulated surface states characteristic of the ‘stacked spin spiral’ (this is just a semantic issue, I know what the authors mean, but the nomenclature might be confusing).

Note that all page numbers and references without special instructions refer to the newly revised manuscript attached with this response and not to the original version submitted. The added contents are marked in red in the revised manuscript.

Response to Reviewer #1:

Reviewer #1: I would like to recommend its publication in NC since the authors have addressed all my comments properly. And I believe that the results presented in this work can enrich the research area of skyrmionics and spintronics.

Response: We greatly appreciate your support in recommending its publication in Nature Communications.

Response to Reviewer #2:

Reviewer #2: I have carefully read replies from the authors to three reviewers including me and the revised manuscript. As a result, I found that the authors have sincerely addressed all the comments, questions and criticisms raised by the three reviewers.

In my previous report, I criticized that the demonstrated skyrmion-bundle generation is far from the original purpose of this experiment which aimed to create stable skyrmion bundles at zero magnetic field. Actually, when I read the original manuscript, I thought that the generated skyrmion bundles survive only under application of magnetic field after reversing the magnetic field. However, after reading the authors reply to my criticism, I realized that I misunderstood the complicated procedure of the skyrmion-bundle generation and that the skyrmion bundles survive even at zero magnetic field.

I think that this aspect was not clearly presented in the previous manuscript. The authors added new conceptual figures (Fig.1) to explain their scenario of creating skyrmion bundles as well as detailed explanations of the procedure in the manuscript. As a result, the advancement of their proposal is now clearly presented.

As for my questions about possible technical application of skyrmion bundles and new important physics contained in the present work, the authors provided convincing answers and revised the manuscript and the supplemental information by adding new argument, figures and references. I found them of satisfactory level. I agree that the experimental demonstration of skyrmion bundles stable at zero field and room temperature without any geometrical confinement is very important for the research on the topological magnetism. Furthermore, the concept of fractional hopfions in the skyrmion bundles explained with a newly added figure (Fig.6) is interesting and important for the fundamental science.

After the entire revisions by the authors, I think that the advancement of the proposed method and the important new physics lying in the experimental results are clearly presented in this paper. After careful consideration, I have changed my previous mind against publication of this work in Nature Communications and now recommend its publication.

Response: We are very happy to obtain your recommendation for our manuscript's publication in Nature Communications. We would like to seize this opportunity once again to express our profound gratitude for your invaluable support.

Response to Reviewer #3:

Reviewer #3: The authors have answered most of my original questions. They have improved the manuscript in several ways, for example by including a new Fig. 1 to better explain the 3D structure of the skyrmion bundles, and by including new simulations about possible fractional hopfion states.

However, for me, this still does not help sufficiently distinguish the work from their previous report on FeGe. From what I can see, the other referees agree that the quality of the data and experimental practices is not in doubt – they are both excellent. Therefore, the question comes down to the novelty of the main conclusions relative to published works.

Response: We greatly appreciate your time and efforts in reviewing our manuscript. We extend our heartfelt thanks for largely recognizing the efforts made within our revised manuscript. With these improvements, Reviewer #1 believes our work “*can enrich the research area of skyrmionics and spintronics*” and Reviewer #2 thinks our work “*is very important for the research on the topological magnetism and is interesting and important for the fundamental science*”. We sincerely request you to reconsider our submission for publication in Nature Communications.

The authors have three main conclusions, which I evaluate below:

1) The observation of skyrmion bundles at room temperature and zero applied magnetic field:

This is well-evidenced throughout the paper, and is the best claim of novelty. I think the comments of referee #2 miss the mark here: yes, the skyrmion bundles are not generated by a current pulse at zero field directly, but following the field manipulation procedure, the skyrmion bundles do survive at zero field at room temperature.

Response: We thank you very much for supporting this claim of novelty. Here, we must express our great respect for your previous comment about adding a scenario figure, which shows the creation of skyrmion bundles at room temperature and zero field. We thank you for pointing out the miss of Reviewer #2. This scenario helps the clarity of our work and enables the support from Reviewer #2.

2) The ability to nucleate skyrmion bundles without field-cooling, instead by applying a current pulse:

The authors create the precursor states for the skyrmion bundles by applying a current pulse. As the authors themselves admit, this is in reality a Joule heating effect (indeed, both of the references the authors’ cite about this point, references 29 and 36, are about Joule heating nucleation of skyrmions). Thus, the authors actually are field-cooling their sample. They are just doing so with a local Joule heating effect. And importantly, they cannot nucleate the skyrmion bundles directly. I think their nucleation method still necessitates the Joule heating.

Response: We agree with you that our method necessitates Joule heating, which has been demonstrated in our manuscript, see Lines 147-148: “*This creation process can be understood by the combined current effect of spin transfer torque and Joule thermal heating*” and Lines 149-152: “*Because the current density is inversely proportional to the thickness, skyrmions are first created at the thin region because of larger temperature increases induced by the current. Then, the spin transfer torque could drive skyrmions nucleated in the thin region to the thick region*”. Here, we propose that this is technically different from the previous creation process by the combination of the current method.

3) The mapping of a magnetic phase diagram for the skyrmion bundle stability: This claim is clearly demonstrated, as evidenced in Fig. 5. However, similarly to referee #2, I don’t know if we should be surprised by this data. It is in keeping with my expectation that the skyrmions would only be stable for small applied fields, since the internal skyrmions are inherently energetically unfavourable in the oppositely applied magnetic field.

Response: We believe that the phase diagram for the skyrmion bundle stability reveals important data, *i.e.* isolated skyrmion bundles can survive at zero magnetic fields in a broad temperature range in free geometries. Such a zero-field stability of skyrmion bundles is understood by the presence of helix pinned along the depth orientation. This part is totally new and far beyond our previous results published in Nature Nanotechnology. Furthermore, as is agreed with other Reviewers, the room-temperature and zero-field stability of skyrmion bundles realized in our manuscript “*is very important for the research on the topological magnetism*”, because both room-temperature and zero-field are the prerequisite for topological spintronic applications.

The authors have added an additional section concerning fractional hopfions and hopfions (presumably in reaction to two recent publications [X. Z. Yu et al. Advanced Materials 35, 2210646 (2023), F. Zheng, et al. Nature 623, 718-723 (2023)]). I agree with the authors that their skyrmion bundles can probably be related to both of these topological states, and this is an interesting point of discussion (actually, maybe the skyrmion bundles are quite similar to the hopfion rings claimed in the latter reference). However, at the moment this section feels very disjointed by the way which the authors have squeezed it into the end of the manuscript, and no reference to this part is made in the abstract, introduction or conclusions. I’m also not convinced it presents a significant advance in the understanding of these 3D hopfion-like states, in the present form.

Following this evaluation, I am still not convinced that the work warrants publication in Nature Communications. The authors present multiple incremental advances on the topic of these interesting 3D spin textures, but I believe these advances are within the expectations of the work they presented in their previous Nature Nanotechnology on skyrmion bundles in FeGe, and the two recent hopfion publications. At the moment, I cannot recommend publication.

I think the authors are in the position to answer several key questions about these 3D spin textures: namely, the detailed 3D structure and topology of the magnetic states, and what is/isn't distinguishing these skyrmion bundles from hopfion states. However, for the moment these questions are not directly addressed in the present manuscript, and it is quite a different topic to what the authors have chosen to focus on (namely, zero field, room temperature skyrmion bundles).

Response: We thank you for evaluating our added Hopfion section “*an interesting point of discussion*”. As agreed by Reviewer #2, this section “*is interesting and important for the fundamental science*”, because our manuscript first reveals that skyrmion bundles could be a natural platform for exploiting diverse magnetic Hopfions, which is not referred to in the suggested Hopfion publications [Advanced Materials 35, 2210646 (2023) and Nature 623, 718-723 (2023)].

We agree on the importance of 3D topological spin textures. The simulation results highly coincide with the experiments, which enable us to further explore the exact structures and topology of our observations by simulations. The simulations reveal that our experimental contrasts correspond to skyrmion bundles, which contain a fractional Hopfion for any topological charges.

Following your suggestion “*no reference to this part is made in the abstract, introduction or conclusions*”, we have referred to the topological nature of skyrmion bundles in the abstract, introduction, and conclusion sections of our revised manuscript.

See the abstract section in Lines 32-34: “Our experimental findings are consistently corroborated by micromagnetic simulations, *which reveal the topological magnetic nature of a skyrmion bundle as skyrmion tubes encircled by a fractional Hopfion.*”

See the Introduction part in Lines 73-74: “*Finally, we further elucidated the connection between skyrmion tubes and magnetic Hopfions*³³⁻³⁶.”

See the conclusion section in Lines 335-337: “*Moreover, we have clarified the topological nature of the boundary spiral of skyrmion bundles as magnetic fractional Hopfions, which suggests the diverse topological magnetism for skyrmion bundles.*”

We believe our proposed room-temperature and zero-field stability of isolated skyrmion bundles in free geometries, as well as the topological connection between skyrmion bundles and fractional Hopfions are all not referred to in previous publications. These observations could promote the development of both topology-based spintronic devices and fundamental topological sciences. Here, we sincerely request you to consider our manuscripts published in Nature Communications.

I give a few more detailed comments below:

Response: Below please find our response to three technical points.

Comment 1: The new Figure 1 is much cleaner, and now I understand a little better the 3D structure of the skyrmion bundles. It's very interesting that the top and bottom layers

have a very different topological configuration compared to the middle layers, despite having the same $Q=2$ topological charge. At the moment, the description of Fig. 1 in the main text is still lacking, and I was still confused about what the surface chiral state was – it looks like two antiskyrmion-like textures are the objects contributing to the topological charge. I found later on in the manuscript that this description is indeed correct, when the authors make the same statement. The authors might consider moving this description to the section describing the Fig. 1.

Response: We thank you for your careful reading. Following your nice suggestions, we have moved the description to the section describing Fig. 1. See Lines 83-90: “*Fig. 1 contains a representative simulated 3D magnetic configuration of skyrmion bundles containing 3 skyrmions. The iso-surfaces correspond to a value of -0.1 for the normalized out-of-plane magnetization component, i.e., $m_z = -0.1$ (Fig. 1b and 1d). When approaching the sample surfaces (Fig. 1e and 1g), where the magnetic vortex is a bi-antiskyrmion with $Q = 2$, and the complete skyrmion bags with $Q = 2$ are located only in the middle layers (Fig. 1f). Therefore, despite the strong spin twist along the depth dimension due to the conical background magnetizations, topological charges maintain $Q = 2$ throughout all layers (Supplemental Fig. S1 and Movie 1)*”.

We have also specified the configurations in surfaces as antiskyrmions in Fig. 1, as shown in Fig. R1.

Fig. R1. (i.e. Fig. 1b-g in the main text)

Comment 2: As mentioned, the configuration of the surface and middle states are very different, although the topological charge remains $Q=2$. I think this could be very interesting, because it means that there should be some complex modulation along the vertical axis. Maybe even Bloch points or magnetic monopoles somewhere in the 3D structure, in order to convert between these magnetic structures along the vertical axis. I am interested if this is true, or whether it really is a smooth deformation between these two very different configurations. This reminds me of recent works on the quadrupole Bloch point configuration in antiskyrmions – one of which was authored by some of the present authors [F. Yasin et al. *Advanced Materials* 2311737 (2024), J. Tang et al.

Science Bulletin 68, 2919-2923]. Could there perhaps be similar interesting 3D structure, if the authors take a closer look into the details of the simulated skyrmion bundles. How does the magnetic configuration change so much through the vertical thickness? Or is it really a smooth change between two configurations of $Q=2$ topology?

Response: We thank you for your constructive comments. We have added a supplementary figure S1 (as shown in Fig. R2) and a Movie 1 to show the magnetic modulation through all layers. It is truly a smooth change between $Q = 2$ bi-antiskyrmions in the surface and $Q = 2$ skyrmion bags in the interior layer without the mediation of Bloch points or monopoles. I think these results well coincide with the basic concept of topology: the properties of space that are preserved under continuous deformations. The referred work [F. Yasin et al. Advanced Materials 2311737 (2024), J. Tang et al. Science Bulletin 68, 2919-2923] report the topological reversal between $Q = 1$ antiskyrmions in interior layers and $Q = -1$ skyrmions in surface layers. The Bloch points emerge between such topological reversal, where spins must undergo a sudden reversal to contribute different signs of Q for two adjacent layers.

We have added related discussions in Lines 90-94: “*Noted that the smooth spin modulations between skyrmion bags in interior layers and antiskyrmions in surface layers without topological variations do not necessarily the emergence of Bloch points, which is different from quadrupole of Bloch points sewing skyrmions and antiskyrmions with topological reversal*^{37, 38}.”

And new references:

37. Tang, J. et al. Sewing skyrmion and antiskyrmion by quadrupole of Bloch points. *Sci. Bull.* **68**, 2919-2923 (2023).
38. Yasin, F.S. et al. Bloch point quadrupole constituting hybrid topological strings revealed with electron holographic vector field tomography. *Adv. Mater.*, 2311737 (2024).

Fig. R2 (*i.e.*, Supplemental Fig. S1) Contour of $m_z = -0.1$ and magnetic configurations at different layers of the $Q = 2$ bundles. The colorwheel represents the in-plane magnetizations.

Comment 3: In several places, the authors discuss the zero-field state of their sample as a “stacked spin spiral”. However, this phrase was coined in another context [F. N. Rybakov et al. *New Journal of Physics* 18, 045002 (2016)], and I don’t believe this is what the authors are claiming to have seen in their work. I think in the present work, it is just a typical magnetic helix which is pinned in the out-of-plane direction. The present state is lacking the long-wavelength modulated surface states characteristic of the ‘stacked spin spiral’ (this is just a semantic issue, I know what the authors mean, but the nomenclature might be confusing).

Response: We greatly appreciate your careful reading. Following your nice suggestion to clear this confusion, we have rectified the term “*stacked spin spiral*” in the revised manuscript by replacing it with “*perpendicular helix*”, and explaining the perpendicular helix as the helix which is pinning in the out-of-plane direction.

See lines 244-246: “Here, we show that isolated zero-field skyrmion bundles can be created in the *perpendicular helix*, whose spins keep uniform within each plane and modulate along the depth to form helix (Supplemental Figure S8), without the application of geometrical confinement effects”.

Reviewers' Comments:

Reviewer #3:

Remarks to the Author:

The authors have answered my lingering questions. I think the comparison of these bundle states to the recently published hopfion studies is useful, and will compel further work on these new 3D topological states. I can recommend the article for publication.

Manuscript ID: NCOMMS-23-56247B

"Stable Skyrmion Bundles at Room Temperature and Zero Magnetic Field in a Chiral Magnet" by Y. Zhang et al.

Note that all page numbers and references without special instructions refer to the newly revised manuscript attached with this response and not to the original version submitted.

Response to Reviewer #3:

Reviewer #3: The authors have answered my lingering questions. I think the comparison of these bundle states to the recently published hopfion studies is useful, and will compel further work on these new 3D topological states. I can recommend the article for publication.

Response: We greatly appreciate your support in recommending its publication in *Nature Communications*.